# ∞-DIFF: INFINITE RESOLUTION DIFFUSION WITH SUBSAMPLED MOLLIFIED STATES

**Sam Bond-Taylor, Chris G. Willcocks**
Department of Computer Science
Durham University
{samuel.e.bond-taylor, christopher.g.willcocks}@durham.ac.uk

## ABSTRACT

This paper introduces ∞-Diff, a generative diffusion model defined in an infinite-dimensional Hilbert space, which can model infinite resolution data. By training on randomly sampled subsets of coordinates and denoising content only at those locations, we learn a continuous function for arbitrary resolution sampling. Unlike prior neural field-based infinite-dimensional models, which use point-wise functions requiring latent compression, our method employs non-local integral operators to map between Hilbert spaces, allowing spatial context aggregation. This is achieved with an efficient multi-scale function-space architecture that operates directly on raw sparse coordinates, coupled with a mollified diffusion process that smooths out irregularities. Through experiments on high-resolution datasets, we found that even at an $8\times$ subsampling rate, our model retains high-quality diffusion. This leads to significant run-time and memory savings, delivers samples with lower FID scores, and scales beyond the training resolution while retaining detail.

## 1 INTRODUCTION

Denoising diffusion models (Song and Ermon, 2019; Ho et al., 2020) have become a dominant choice for data generation, offering stable training and the ability to generate diverse and high quality samples. These methods function by defining a forward diffusion process which gradually destroys information by adding Gaussian noise, with a neural network then trained to denoise the data, in turn approximating the data distribution. Scaling diffusion models to higher resolutions has been the topic of various recent research, with approaches including iteratively upsampling lower resolution images (Ho et al., 2022a) and operating in a compressed latent space (Rombach et al., 2022).

Deep neural networks typically assume that data can be represented with a fixed uniform grid, however, the underlying signal is often continuous. As such, these approaches scale poorly with resolution. Neural fields (Xie et al., 2022; Sitzmann et al., 2020; Mildenhall et al., 2021) address this problem by directly representing data as a mapping from coordinates to intensities (such as pixel values), making the parameterisation and memory/run-time from resolution independent from resolution, thereby allowing training on data that would not usually fit in memory. Neural field-based generative models (Dupont et al., 2022a;b; Bond-Taylor and Willcocks, 2021; Du et al., 2021) have been developed to take advantage of these properties. Being inherently independent between coordinates, these

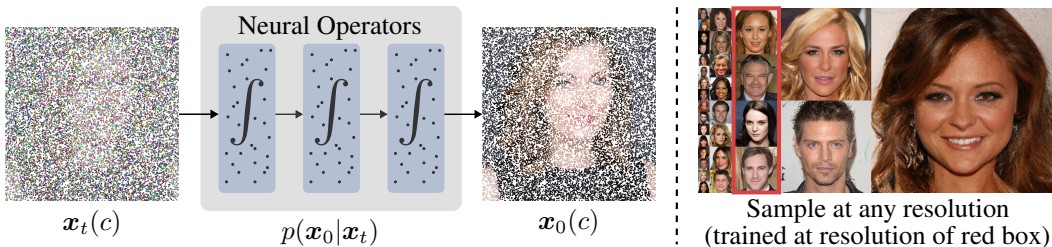

$$\boldsymbol{x}_t(c) \qquad p(\boldsymbol{x}_0|\boldsymbol{x}_t) \qquad \boldsymbol{x}_0(c) \qquad \text{Sample at any resolution} \atop \text{(trained at resolution of red box)}$$

Figure 1: We define a diffusion process in an infinite dimensional image space by randomly sampling coordinates and training a model parameterised by neural operators to denoise at those coordinates.

---

Source code is available at https://github.com/samb-t/infty-diff.

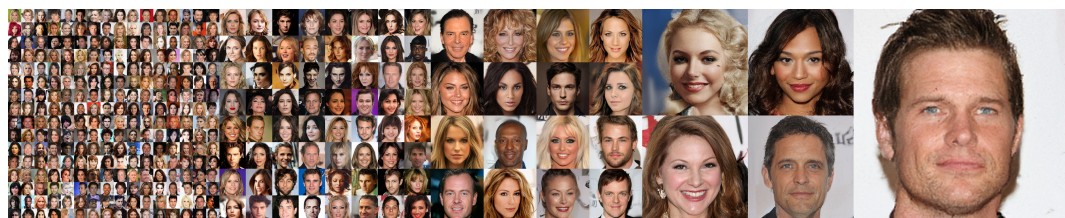

Figure 2: Modelling data as functions allows sampling at arbitrary resolutions using the same model with different sized noise. Left to right: 64×64, 128×128, 256×256 (original), 512×512, 1024×1024.

approaches condition networks on compressed latent vectors to provide global information. However, the sample quality of these methods is significantly lower than finite-dimensional generative models.

In this work we develop an approach that substantially improves upon the quality and scaling of existing infinite dimensional generative models, reducing the gap to finite-dimensional methods, while retaining the benefits of infinite dimensional models: subsampling coordinates to decouple memory/run-time from resolution, making scaling more computationally feasible, while also allowing training and sampling at arbitrary resolutions. We achieve this by designing a Gaussian diffusion model in an infinite dimensional state space[1]. We argue that compressed latent-based neural fields cannot effectively be used to parameterise such diffusion models due to the reliance on compression, going against standard diffusion architecture design, with it also being impractical to compress states to latents at every step. Instead, we propose using non-local integral operators to model the denoising function, aggregating both global and local information in order to effectively denoise the data.

Specifically, we propose $\infty$-Diff, addressing these issues:

- We introduce a Gaussian diffusion model defined in an infinite-dimensional state space that allows complex arbitrary resolution data to be generated (see Fig. 2)[1].
- We design a powerful and scalable, function-space architecture that operates directly on raw sparsely subsampled coordinates, enabling improvements in run-time and memory usage.
- We achieve state-of-the-art FID scores on multiple high-res image datasets, trained with up to $8\times$ subsampling, substantially outperforming prior infinite resolution generative models.

## 2 BACKGROUND

Here we review Gaussian diffusion models (Section 2.1) and generative neural fields (Section 2.2).

### 2.1 DIFFUSION MODELS

Gaussian diffusion models (Sohl-Dickstein et al., 2015; Ho et al., 2020) are formed by defining a forward process $q(\boldsymbol{x}_{1:T}|\boldsymbol{x}_0)$ that gradually adds noise to the data, $\boldsymbol{x}_0 \sim q(\boldsymbol{x}_0)$, over $T$ steps, resulting in a sequence of latent variables $\boldsymbol{x}_1, \ldots, \boldsymbol{x}_T$ such that $q(\boldsymbol{x}_T) \approx \mathcal{N}(\boldsymbol{x}_T; \mathbf{0}, \boldsymbol{I})$. The reverse of this process can also be expressed as a Markov chain $p(\boldsymbol{x}_{0:T})$. Choosing Gaussian transition densities chosen to ensure these properties hold, the densities may be expressed as

$$q(\boldsymbol{x}_{1:T}|\boldsymbol{x}_0) = \prod_{t=1}^{T} q(\boldsymbol{x}_t|\boldsymbol{x}_{t-1}), \qquad q(\boldsymbol{x}_t|\boldsymbol{x}_{t-1}) = \mathcal{N}(\boldsymbol{x}_t; \sqrt{1-\beta_t}\boldsymbol{x}_{t-1}, \beta_t\boldsymbol{I}), \qquad (1)$$

$$p(\boldsymbol{x}_{0:T}) = p(\boldsymbol{x}_T) \prod_{t=1}^{T} p(\boldsymbol{x}_{t-1}|\boldsymbol{x}_t), \qquad p(\boldsymbol{x}_{t-1}|\boldsymbol{x}_t) = \mathcal{N}(\boldsymbol{x}_{t-1}; \boldsymbol{\mu}_\theta(\boldsymbol{x}_t, t), \boldsymbol{\Sigma}_\theta(\boldsymbol{x}_t, t)), \qquad (2)$$

where $0 < \beta_1, \ldots, \beta_T < 1$ is a pre-defined variance schedule and the covariance is typically of the form $\boldsymbol{\Sigma}_\theta(\boldsymbol{x}_t, t) = \sigma_t^2 \boldsymbol{I}$ where often $\sigma_t^2 = \beta_t$. Aiding training efficiency, $q(\boldsymbol{x}_t|\boldsymbol{x}_0)$ can be expressed in closed form as $q(\boldsymbol{x}_t|\boldsymbol{x}_0) = \mathcal{N}(\boldsymbol{x}_t; \sqrt{\bar{\alpha}_t}\boldsymbol{x}_0, (1-\bar{\alpha}_t)\boldsymbol{I})$ where $\bar{\alpha}_t = \prod_{s=1}^{t} \alpha_s$ for $\alpha_s = 1 - \beta_s$. Training is possible by optimising the evidence lower bound on the negative log-likelihood, expressed as the KL-divergence between the forward process posteriors and backward transitions at each step

$$\mathcal{L} = \sum_{t \geq 1} \mathbb{E}_q\left[D_{\mathrm{KL}}(q(\boldsymbol{x}_{t-1}|\boldsymbol{x}_t, \boldsymbol{x}_0) \| p(\boldsymbol{x}_{t-1}|\boldsymbol{x}_t))\right] = \sum_{t \geq 1} \mathbb{E}_q\left[\frac{1}{2\sigma_t^2}\|\tilde{\boldsymbol{\mu}}_t(\boldsymbol{x}_t, \boldsymbol{x}_0) - \boldsymbol{\mu}_\theta(\boldsymbol{x}_t, t)\|_2^2\right], \quad (3)$$

---

[1]A number of parallel works including Kerrigan et al. (2023); Lim et al. (2023); and Franzese et al. (2023) also developed infinite-dimensional diffusion models that use non-local integral operators, see Sec. 6 for more.

for $q(\boldsymbol{x}_{t-1}|\boldsymbol{x}_t, \boldsymbol{x}_0) = \mathcal{N}(\boldsymbol{x}_{t-1}; \tilde{\boldsymbol{\mu}}_t(\boldsymbol{x}_t, \boldsymbol{x}_0), \tilde{\beta}_t \boldsymbol{I})$, where $\tilde{\boldsymbol{\mu}}_t$ and $\tilde{\beta}_t$ can be derived in closed form. It is typical to reparameterise $\boldsymbol{\mu}_\theta$ to simplify the variational bound, one example of such found to improve visual quality is $\boldsymbol{\mu}_\theta(\boldsymbol{x}_t, t) = \frac{1}{\alpha_t}(\boldsymbol{x}_t - \frac{\beta_t}{\sqrt{1-\bar{\alpha}_t}}\boldsymbol{\epsilon}_\theta(\boldsymbol{x}_t, t))$, where the denoising network instead predicts the added noise. Diffusion models are closely connected with score-matching models; this can be seen by making the approximation (De Bortoli et al., 2021),

$$p(\boldsymbol{x}_{t-1}|\boldsymbol{x}_t) = p(\boldsymbol{x}_t|\boldsymbol{x}_{t-1}) \exp(\log p(\boldsymbol{x}_{t-1}) - \log p(\boldsymbol{x}_t)) \tag{4}$$

$$\approx \mathcal{N}(\boldsymbol{x}_{t-1}; \sqrt{1-\beta_t}\boldsymbol{x}_t + \beta_t \nabla_{\boldsymbol{x}_t} \log p(\boldsymbol{x}_t), \beta_t \boldsymbol{I}), \tag{5}$$

which holds for small values of $\beta_t$. While $\nabla_{\boldsymbol{x}_t} \log p(\boldsymbol{x}_t)$ is not available, it can be approximated using denoising score matching methods (Hyvärinen, 2005; Vincent, 2011). Given that $\nabla_{\boldsymbol{x}_t} \log p(\boldsymbol{x}_t) = \mathbb{E}_{p(\boldsymbol{x}_0|\boldsymbol{x}_t)}[\nabla_{\boldsymbol{x}_t} \log p(\boldsymbol{x}_t|\boldsymbol{x}_0)]$ we can learn an approximation to the score with a neural network parameterised by $\theta$, $s_\theta(\boldsymbol{x}_t, t) \approx \nabla \log p(\boldsymbol{x}_t)$ (Song and Ermon, 2019), by minimising a reweighted variant of the ELBO (Eq. 3). One problem with diffusion models is the slow sequential sampling; to speed this up, denoising diffusion implicit models (DDIMs) transform diffusion models into deterministic models allowing fewer steps to yield the same quality, replacing the sampling steps with

$$\boldsymbol{x}_{t-1} = \sqrt{\alpha_{t-1}}\left(\frac{\boldsymbol{x}_t - \sqrt{1-\alpha_t}\boldsymbol{\epsilon}_\theta(\boldsymbol{x}_t, t)}{\sqrt{\alpha_t}}\right) + \sqrt{1-\alpha_{t-1}} \cdot \boldsymbol{\epsilon}_\theta(\boldsymbol{x}_t, t). \tag{6}$$

## 2.2 GENERATIVE NEURAL FIELDS

Neural fields (Xie et al., 2022; Mildenhall et al., 2021; Sitzmann et al., 2020) are an approach for continuous data representation that map from coordinates $\boldsymbol{c}$ to values $\boldsymbol{v}$ (such as RGB intensities), $f_\theta(\boldsymbol{c}) = \boldsymbol{v}$. This decouples the memory needed to represent data from its resolution. The mapping function, $f_\theta$, typically an MLP network, is optimised by minimising a reconstruction loss with ground truth values at each coordinate. The local nature of $f_\theta$ allows the loss to be Monte-Carlo approximated by evaluating $f_\theta$ on subsets of coordinates, allowing higher resolution data than would fit in memory to be trained on. Since $f_\theta$ is independent per coordinate, being unable to transform over multiple points like convolutions/transformers, to represent spaces of functions approaches generally also condition on compressed latent vectors $\boldsymbol{z}$ used to describe single data points, $f_\theta(\boldsymbol{c}, \boldsymbol{z})$. Dupont et al. (2022a) first uses meta-learning to compress the dataset into latent conditional neural fields, then approximates the distribution of latents with a DDPM (Ho et al., 2020) or Normalizing Flow (Rezende and Mohamed, 2015). Bond-Taylor and Willcocks (2021) form a VAE-like model with a single gradient step to obtain latents. Zhuang et al. (2023) design a diffusion model with a small subset of coordinates used to provide context. Finally, some approaches use hypernetworks to output the weight of neural fields: Dupont et al. (2022b) define the hypernetwork as a generator in an adversarial framework, and Du et al. (2021) use manifold learning to represent the latent space.

## 3 INFINITE DIMENSIONAL DIFFUSION MODELS

In this section we extend diffusion models to infinite dimensions in order to allow higher-resolution data to be trained on by subsampling coordinates during training and permit training/sampling at arbitrary resolutions. We argue that application of conditional neural fields to diffusion models is problematic due to the need to compress to a latent vector, adding complexity and opportunity for error, instead, the denoising function should be a non-local integral operator with no compression. A number of parallel works also developed diffusion models in infinite dimensions, including Kerrigan et al. (2023); Lim et al. (2023); and Franzese et al. (2023); we recommend also reading these works, which go further in the theoretical treatment, while ours focuses more on design and practical scaling.

To achieve this, we restrict the diffusion state space to a Hilbert space $\mathcal{H}$, elements of which, $x \in \mathcal{H}$, are functions, e.g. $x: \mathbb{R}^n \to \mathbb{R}^d$. Hilbert spaces are equipped with an inner product $\langle \cdot, \cdot \rangle$ and corresponding norm $\|\cdot\|_\mathcal{H}$. For simplicity we consider the case where $\mathcal{H}$ is the space of $L^2$ functions from $[0, 1]^n$ to $\mathbb{R}^d$ although the following sections can be applied to other spaces. As such, a point in $\mathcal{H}$ could represent an image, audio signal, video, 3D model, etc. A Gaussian measure $\mu$ can be defined in $\mathcal{H}$ in terms of its characteristic function $\hat{\mu}$ (Da Prato and Zabczyk, 2014),

$$\hat{\mu}(x) = \exp\left(i\langle x, m \rangle + \tfrac{1}{2}\langle Cx, x \rangle\right), \tag{7}$$

where the mean $m$ lies in $\mathcal{H}$, $m \in \mathcal{H}$ and the covariance operator ($C : \mathcal{H} \to \mathcal{H}$) is self-adjoint (denoted $C = C^*$), non-negative (i.e. $C \geq 0$), and trace-class ($\int_\mathcal{H} \|x\|_\mathcal{H} d\mu(x) = \text{tr}(C) < \infty$)

White Noise Diffusion

Mollified Diffusion

Figure 3: Example diffusion processes. Mollified diffusion smooths diffusion states allowing the space to be more effectively modelled with continuous operators.

(Kukush, 2020). For a Gaussian random element $x$ with distribution $\mu$, $x \sim \mathcal{N}(m, C)$. The Radon-Nikodym theorem states the existence of a density for a measure $v$ absolutely continuous with respect to a base measure $\mu$: for example, the density between two Gaussians is given by Minh (2021); see Lim et al. (2023); Kerrigan et al. (2023) for more detail in the context of functional diffusion models.

### 3.1 MOLLIFICATION

When defining diffusion in infinite dimensions, it may seem natural to use white noise in the forwards process, where each coordinate is an independent and identically distributed Gaussian random variable; that is, $\mathcal{N}(0, C_I)$ where $C_I(z(s), z(s')) = \delta(s - s')$, using the Dirac delta function $\delta$. However, this noise does not lie in $\mathcal{H}$ (Da Prato and Zabczyk, 2014) with it not satisfying the trace-class requirement. Instead, obtain Gaussian noise in $\mathcal{H}$ by convolving white noise with a mollifier kernel $k(s) > 0$ corresponding to a linear operator $T$, giving $\mathcal{N}(0, TT^*)$, smoothing the white noise to lie in $\mathcal{H}$ (Higdon, 2002). To ensure one-to-one correspondence between kernel and noise, $k$ must satisfy $\int_{\mathbb{R}^d} k(s)ds < \infty$ and $\int_{\mathbb{R}^d} k^2(s)ds < \infty$, making $TT^*$ self-adjoint and non-negative. Considering $k$ to be a Gaussian kernel with smoothing parameter $l > 0$, $h = Tx$ is given by

$$h(c) = \int_{\mathbb{R}^n} K(c - y, l)x(y)\,\mathrm{d}y, \text{ where } K(y, l) = \frac{1}{(4\pi l)^{\frac{n}{2}}}e^{-\frac{|y|^2}{4l}}. \tag{8}$$

### 3.2 INFINITE DIMENSIONAL MOLLIFIED DIFFUSION

To formulate a diffusion model in $\mathcal{H}$, we must specify the transition distributions. However, irregularity in data points $x$ can impact stability, leading to the model being unable to generalise across different subsampling rates/resolutions. This can be mitigated by careful hyperparameter tuning or, in our case, by also mollifying $x$ (as with the previous noise mollification). While the necessity of this depends on the nature of $x$, we have included it for completeness. First, we define the marginals

$$q(x_t|x_0) = \mathcal{N}(x_t; \sqrt{\bar{\alpha}_t}Tx_0, (1 - \bar{\alpha}_t)TT^*), \tag{9}$$

where coefficients are the same as in Section 2.1. From this we are able to derive a closed form representation of the posterior (proof in Appendix B.1),

$$q(x_{t-1}|x_t, x_0) = \mathcal{N}(x_{t-1}; \tilde{\mu}_t(x_t, x_0), \tilde{\beta}_t TT^*),$$

$$\text{where } \quad \tilde{\mu}_t(x_t, x_0) = \frac{\sqrt{\bar{\alpha}_{t-1}}\beta_t}{1 - \bar{\alpha}_t}Tx_0 + \frac{\sqrt{\alpha_t}(1 - \bar{\alpha}_{t-1})}{1 - \bar{\alpha}_t}x_t \quad \text{and} \quad \tilde{\beta}_t = \frac{1 - \bar{\alpha}_{t-1}}{1 - \bar{\alpha}_t}\beta_t. \tag{10}$$

Defining the reverse transitions as $p_\theta(x_{t-1}|x_t) = \mathcal{N}(x_{t-1}; \mu_\theta(x_t, t), \sigma_t^2 TT^*)$, then we can parameterise $\mu_\theta : \mathcal{H} \times \mathbb{R} \to \mathcal{H}$ to directly predict $x_0$. The loss in Eq. (3) can be extended to infinite dimensions (Pinski et al., 2015). However, since Ho et al. (2020) find that predicting the noise yields higher image quality, we parameterise $\mu_\theta$ to predict $\xi \sim \mathcal{N}(0, TT^*)$, motivated by the rewriting the loss as

$$\mathcal{L}_{t-1} = \mathbb{E}_q\left[\frac{1}{2\sigma_t^2}\left\|T^{-1}\left(\frac{1}{\sqrt{\alpha_t}}\left(x_t(x_0, \xi) - \frac{\beta_t}{\sqrt{1 - \bar{\alpha}_t}}\xi\right) - \mu_\theta(x_t, t)\right)\right\|_{\mathcal{H}}^2\right], \tag{11}$$

$$\mu_\theta(x_t, t) = \frac{1}{\sqrt{\alpha_t}}\left[x_t - \frac{\beta_t}{\sqrt{1 - \bar{\alpha}_t}}f_\theta(x_t, t)\right], \tag{12}$$

where $x_t(x_0, \xi) = \sqrt{\bar{\alpha}_t}x_0 + \sqrt{1 - \bar{\alpha}_t}\xi$. Since $T^{-1}$ does not affect the minima, we follow Ho et al. (2020) and use a simplified loss, $\mathcal{L}_{t-1}^{\text{simple}} = \mathbb{E}_q[\|f_\theta(x_t, t) - \xi\|_{\mathcal{H}}^2]$. The concurrent work by Kerrigan

et al. (2023) showed that in the infinite-dimensional limit, the loss will be finite only for specific choices of $\tilde{\beta}_t$, while Lim et al. (2023) found similar only for specific parameterisations of $\mu_\theta$; however, since the loss is Monte-Carlo approximated, this is not problematic in practice.

**Data Mollification** By mollifying the training data $x_0$ to ensure regularity, resulting samples are similarly regular; directly predicting $x_0$ would give an estimate of the original data, but by predicting $\xi$ we are only able to sample $Tx_0$. However, in the case of the Gaussian mollifier kernel with adequate boundary conditions, the existence of the inverse $T^{-1}$ is clear if we consider the Fourier transform of $x(c)$, denoted $\hat{x}(\omega)$, then the Gaussian convolution can be defined by $\hat{h}(\omega) = e^{-\omega^2 t}\hat{x}(\omega)$. And so $Tx$ is one-to-one on any class of Fourier transformable functions, with $Tx$ being bounded ensuring uniqueness and therefore invertibility (John, 1955). Explicitly, the inverse is given by $\hat{x}(\omega) = e^{\omega^2 t}\hat{h}(\omega)$ (Hummel et al., 1987). However, inverting is ill-conditioned, with arbitrarily small changes (for instance by floating point error) destroying smoothness (Hummel et al., 1987). In this case, the Wiener filter can for instance be used as an approximate inverse, defined as $\tilde{x}(\omega) = \frac{e^{-\omega^2 t}}{e^{-2(\omega^2 t)}+\epsilon^2}\hat{h}(\omega)$, where $\epsilon$ is an estimate of the inverse SNR (Biemond et al., 1990).

## 4 PARAMETERISING THE DIFFUSION PROCESS

In order to model the denoising function in Hilbert space, there are certain properties that is essential for the class of learnable functions to satisfy so as to allow training on infinite resolution data:

1. Can take as input points positioned at arbitrary coordinates.
2. Generalises to different numbers of input points than trained on, sampled on a regular grid.
3. Able to capture both global and local information.
4. Scales to very large numbers of input points, i.e. efficient in terms of runtime and memory.

Recent diffusion models often use a U-Net (Ronneberger et al., 2015) consisting of a convolutional encoder and decoder with skip-connections between resolutions allowing both global and local information to be efficiently captured. Unfortunately, U-Nets function on a fixed grid making them unsuitable. However, we can take inspiration to build an architecture satisfying the desired properties.

### 4.1 NEURAL OPERATORS

Neural Operators (Li et al., 2020; Kovachki et al., 2021) are a framework designed for efficiently solving partial differential equations by learning to directly map the PDE parameters to the solution in a single step. However, more generally they are able to learn a map between two infinite dimensional function spaces making them suitable for parameterising an infinite dimensional diffusion model.

Let $\mathcal{X}$ and $\mathcal{S}$ be separable Banach spaces representing the spaces of noisy and denoised data respectively; a neural operator is a map $\mathcal{F}_\theta\colon \mathcal{X} \to \mathcal{S}$. Since $x \in \mathcal{X}$ and $s \in \mathcal{S}$ are both functions, we only have access to pointwise evaluations. Let $\boldsymbol{c} \in \binom{D}{m}$ be an $m$-point discretisation of the domain $D = [0, 1]^n$ (i.e. $\boldsymbol{c}$ is $m$ coordinates), and assume we have observations $x(\boldsymbol{c}) \in \mathbb{R}^{m \times d}$. To be discretisation invariant, the neural operator may be evaluated at any $c \in D$, potentially $c \notin \boldsymbol{c}$, thereby allowing a transfer of solutions between discretisations. Each layer is built using a non-local integral kernel operator, $\mathcal{K}(x; \phi)$, parameterised by neural network $\kappa_\phi$, aggregating information spatially,

$$(\mathcal{K}(x; \phi)v_l)(c) = \int_D \kappa_\phi(c, b, x(c), x(b))v_l(b)\, \mathrm{d}b, \qquad \forall c \in D. \tag{13}$$

Deep networks can be built in a similar manner to conventional methods, by stacking layers of linear operators with non-linear activation functions, $v_0 \mapsto v_1 \mapsto \cdots \mapsto v_L$ where $v_l \mapsto v_{l+1}$ is defined as

$$v_{l+1}(c) = \sigma(Wv_l(c) + (\mathcal{K}(x; \phi)v_l)(c)), \qquad \forall c \in D, \tag{14}$$

for input $v_0 = x$, activations $v_l$, output $v_L = s$, pointwise linear transformation $W\colon \mathbb{R}^d \to \mathbb{R}^d$, and activation function $\sigma\colon \mathbb{R} \to \mathbb{R}$. One example is the Fourier Neural Operator (FNO) (Li et al., 2021),

$$(\mathcal{K}(x; \phi)v_l)(c) = \mathcal{G}^{-1}\left(R_\phi \cdot (\mathcal{G}v_t)\right)(c), \qquad \forall c \in D, \tag{15}$$

where $\mathcal{G}$ is the Fourier transform, and $R_\phi$ is learned transformation in Fourier space. When coordinates lie on a regular grid, the fast Fourier transform can be used, making FNOs fast and scalable.

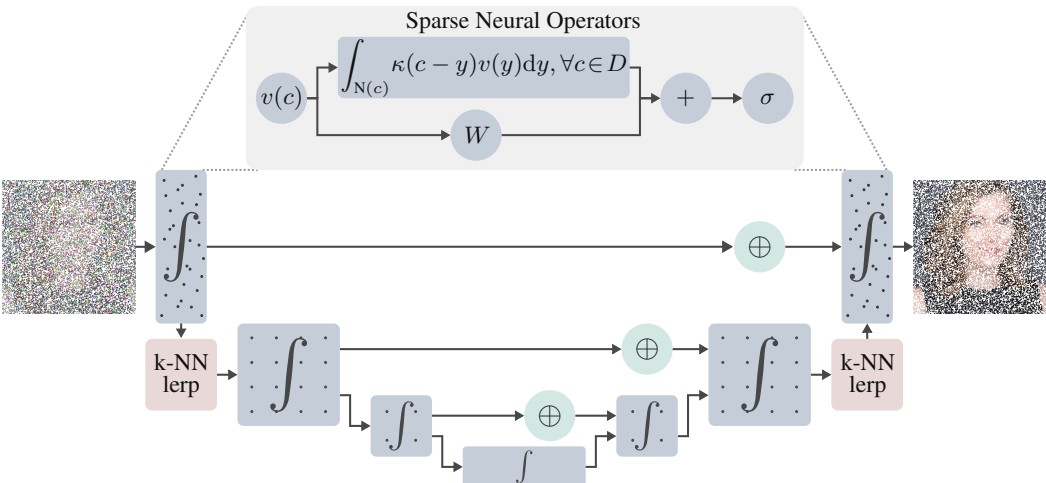

Figure 4: ∞-Diff uses a hierarchical architecture that operates on irregularly sampled functions at the top level to efficiently capture fine details, and on fixed grids at the other levels to capture global structure. This approach allows scaling to intricate high-resolution data.

## 4.2 MULTI-SCALE ARCHITECTURE

While neural operators which satisfy all required properties (1-4) exist, such as Galerkin attention (Cao, 2021) (softmax-free linear attention) and MLP-Mixers (Tolstikhin et al., 2021), scaling beyond small numbers of coordinates is still challenging due to the high memory costs. Instead we design a U-Net inspired multi-scale architecture (Fig. 4) that separately aggregates local/global information.

In a continuous setting, there are two main approaches to downsampling: (1) selecting a subset of coordinates (Wang and Golland, 2022) and (2) interpolating points to a regularly spaced grid (Rahman et al., 2022). We found that with repeated application of (1), approximating integral operators on non-uniformly spaced grids with very few points did not perform nor generalise well, likely due to the high variance. On the other hand, while working with a regular grid removes some sparsity properties, issues with variance are much lower. As such, we use a hybrid approach with sparse operators applied on the raw irregularly sampled data to local regions; after this points are interpolated to a regular grid and a grid-based architecture is applied in order to aggregate global information; if the regular grid is of sufficiently high dimension, this combination should be sufficient. While an FNO (Li et al., 2021; Rahman et al., 2022) architecture could be used, we achieved better results with dense convolutions (Nichol and Dhariwal, 2021), with sparse operators used for resolution changes.

## 4.3 EFFICIENT SPARSE OPERATORS

At the sparse level we use convolution operators (Kovachki et al., 2021), finding them to be more performant than Galerkin attention, with global context provided by the multiscale architecture. This is defined using a translation invariant kernel restricted to the local region of each coordinate, $N(c)$,

$$x(c) = \int_{N(c)} \kappa(c - y)v(y)\,\mathrm{d}y, \qquad \forall c \in D. \tag{16}$$

We restrict $\kappa$ to be a depthwise kernel due to the greater parameter efficiency for large kernels (particularly for continuously parameterised kernels) and finding that they are more able to generalise when trained with fewer sampled coordinates; although the sparsity ratio is the same for regular and depthwise convolutions, because there are substantially more values in a regular kernel, there is more spatial correlation between values. When a very large number of sampled coordinates are used, fully continuous convolutions are extremely impractical in terms of memory usage and run-time. In practice, however, images are obtained and stored on a discrete grid. As such, by treating images as high dimensional, but discrete entities, we can take advantage of efficient sparse convolution libraries (Choy et al., 2019; Contributors, 2022), making memory usage and run-times much more reasonable. Specifically, we use TorchSparse (Tang et al., 2022), modified to allow depthwise convolutions. Wang and Golland (2022) proposed using low discrepancy coordinate sequences to approximate the integrals due to their better convergence rates. However, we found uniformly sampled points to be more effective, likely because high frequency details are able to be more easily captured.

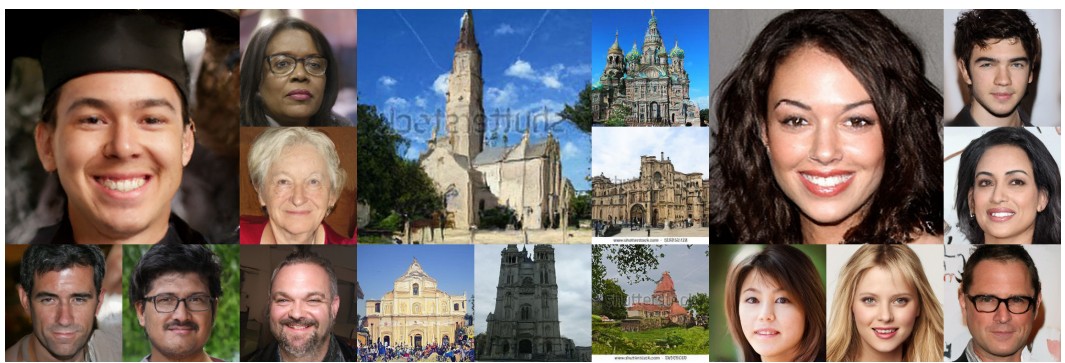

Figure 5: Samples from ∞-Diff models trained on sets of randomly subsampled coordinates.

| Method | CelebAHQ-64 | CelebAHQ-128 | FFHQ-256 | Church-256 |
|---|---|---|---|---|
| **Finite-Dimensional** | | | | |
| CIPS (Anokhin et al., 2021) | - | - | 5.29 | 10.80 |
| StyleSwin (Zhang et al., 2022) | - | 3.39 | 3.25 | 8.28 |
| UT (Bond-Taylor et al., 2022) | - | - | 3.05 | **5.52** |
| StyleGAN2 (Karras et al., 2020) | - | **2.20** | **2.35** | 6.21 |
| **Infinite-Dimensional** | | | | |
| D2F (Dupont et al., 2022a) | 40.4* | - | - | - |
| DPF (Zhuang et al., 2023) | 13.21* | - | - | - |
| GEM (Du et al., 2021) | 14.65 | 23.73 | 35.62 | 87.57 |
| GASP (Dupont et al., 2022b) | 9.29 | 27.31 | 24.37 | 37.46 |
| **∞-Diff (Ours)** | **4.57** | **3.02** | **3.87** | **10.36** |

Table 1: $\text{FID}_{\text{CLIP}}$ (Kynkäänniemi et al., 2023) evaluation against finite-dimensional methods as well as other infinite-dimensional approaches which are trained on coordinate subsets. *=Inception FID.

## 5 EXPERIMENTS

In this section we demonstrate that the proposed mollified diffusion process modelled with neural operator based networks and trained on coordinate subsets are able to generate high quality, high resolution samples. We explore the properties of this approach including discretisation invariance, the impact of the number of coordinates during training, and compare the sample quality of our approach with other infinite dimensional generative models. We train models on $256 \times 256$ datasets, FFHQ (Karras et al., 2019) and LSUN Church (Yu et al., 2015), as well as CelebA-HQ (Karras et al., 2018); unless otherwise specified models are trained on $1/4$ of pixels (to fit in memory), randomly selected.

Very large batch sizes are necessary to train diffusion models due the high variance (Hoogeboom et al., 2023), making training on high resolution data on a single GPU impractical. To address this, we use diffusion autoencoders (Preechakul et al., 2022) to reduce stochasticity: during training, sparsely sampled data are encoded to small latent vectors and used to condition our pixel-level infinite-dimensional diffusion model, allowing better estimation of the denoised data at high time steps. Subsequently, a small diffusion model is quickly trained to model these latents. Our encoder is the downsampling part of our architecture (left half of Fig. 4). When sampling, we use the deterministic DDIM interpretation with 100 steps. Additional details are in Appendix A. Source code is available at https://github.com/samb-t/infty-diff.

**Sample Quality**  Samples from our approach can be found in Fig. 5 which are high quality, diverse, and capture fine details. In Table 1 we quantitatively compare with other approaches that treat inputs as infinite dimensional data, as well as more traditional approaches that assume data lies on a fixed grid. As proposed by Kynkäänniemi et al. (2023), we calculate FID (Heusel et al., 2017) using CLIP features (Radford et al., 2021) which is better correlated with human perception of image quality. Our approach scales to high resolutions much more effectively than the other function-based approaches as evidenced by the substantially lower scores. Visual comparison between samples from our approach and other function-based approaches can be found in Fig. 6 where samples from our approach can be seen to be higher quality and display more details without blurring or adversarial

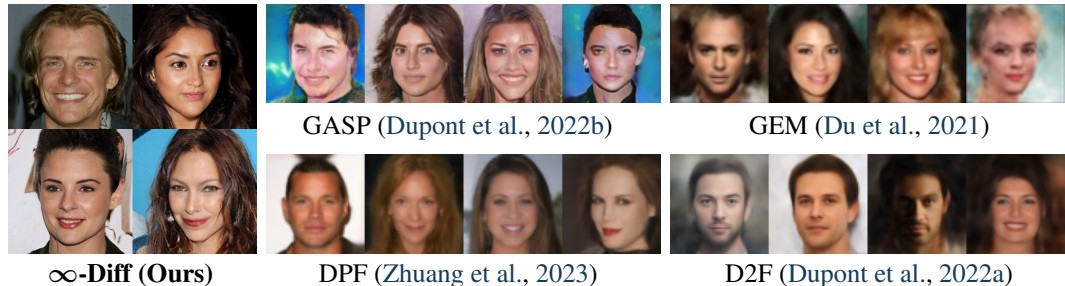

| | |
|---|---|
| GASP (Dupont et al., 2022b) | GEM (Du et al., 2021) |

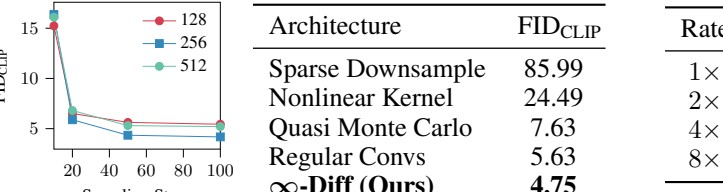

| | |
|---|---|
| ∞-**Diff (Ours)** | DPF (Zhuang et al., 2023) | D2F (Dupont et al., 2022a) |

Figure 6: Qualitative comparison with other infinite dimensional approaches.

| Architecture | FID$_{CLIP}$ |
|---|---|
| Sparse Downsample | 85.99 |
| Nonlinear Kernel | 24.49 |
| Quasi Monte Carlo | 7.63 |
| Regular Convs | 5.63 |
| ∞-**Diff (Ours)** | **4.75** |

| Rate | FID$_{CLIP}$ | Speedup |
|---|---|---|
| $1\times$ | 3.15 | $1.0\times$ |
| $2\times$ | 4.12 | $1.0\times$ |
| $4\times$ | 4.75 | $1.3\times$ |
| $8\times$ | 6.48 | $1.6\times$ |

Figure 7: FID$_{CLIP}$ at various steps & resolutions.

Table 2: Architectural component ablations in terms of FID$_{CLIP}$.

Table 3: Impact of subsampling rate on quality for FFHQ 128. FID$_{CLIP}$ calculated with 10k samples.

artefacts. All of these approaches are based on neural fields (Xie et al., 2022) where coordinates are treated independently; in contrast, our approach uses neural operators to transform functions using spatial context thereby allowing more details to be captured. Both GASP (Dupont et al., 2022b) and GEM (Du et al., 2021) rely on compressed latent-conditional hypernetworks which makes efficiently scaling difficult. D2F (Dupont et al., 2022a) relies on a deterministic compression stage which loses detail due to the finite vector size. DPF (Zhuang et al., 2023) uses small fixed sized coordinate subsets as global context with other coordinates modelled implicitly, thereby causing blur.

**Discretisation Invariance** In Fig. 2 we demonstrate the discretisation invariance properties of our approach. After training on random coordinate subsets from $256\times256$ images, we can sample from this model at arbitrary resolutions which we show at resolutions from $64\times64$ to $1024\times1024$ by initialising the diffusion with different sized noise. We experimented with (alias-free) continuously parameterised kernels (Romero et al., 2022) but found bi-linearly interpolating kernels to be more effective. At each resolution, even exceeding the training data, samples are consistent and diverse. In Fig. 7 we analyse how the number of sampling steps affects quality at different sampling resolutions.

**Coordinate Sparsity** One factor influencing the quality of samples is the number of coordinates sampled during training; fewer coordinates means fewer points from which to approximate each integral. We analyse the impact of this in Table 3, finding that as expected, performance decreases with fewer coordinates, however, this effect is fairly minimal. With fewer coordinates also comes substantial speedup and memory savings; at $256\times256$ with $4\times$ subsampling the speedup is $1.4\times$.

**Architecture Analysis** In Table 2 we ablate the impact of various architecture choices against the architecture described in Section 4.2, matching the architecture as closely as possible. In particular, sparse downsampling (performed by randomly subsampling coordinates; we observed similar with equidistant subsampling, Qi et al., 2017) fails to capture the distribution. Similarly using a spatially nonlinear kernel (Eq. 13), implemented as conv, activation, conv, does not generalise well unlike linear kernels (we observed similar for softmax transformers, Kovachki et al., 2021).

**Super-resolution** The discretisation invariance properties of the proposed approach makes superresolution a natural application. We evaluate this in a simple way, passing a low resolution image through the encoder, then sampling at a higher resolution; see Fig. 8 where it is clear that more details have been added. A downside of this specific approach is that information is lost in the encoding process, however, this could potentially by improved by incorporating DDIM encodings (Song et al., 2021).

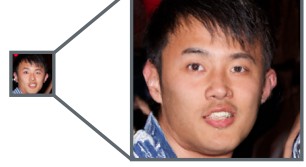

Figure 8: Super-resolution

**Inpainting** Inpainting is possible with mollified diffusion (Fig. 9), using reconstruction guidance (Ho et al., 2022b), $x_{t-1} \leftarrow x_{t-1} - \lambda \nabla_{x_t} \| m \odot (\tilde{\mu}_0(x_t, t) - T\bar{x}) \|_2^2$ for inpainting mask $m$, learned estimate of $Tx_0$, $\tilde{\mu}_0$, and image to be inpainted $\bar{x}$. The diffusion autoencoder framework gives an additional level of control when inpainting since the reverse diffusion process can be applied to encodings from a chosen time step $t_s$, allowing control over how different the inpainted region is from the original image.

Figure 9: Inpainting.

## 6 DISCUSSION

There are a number of interesting directions to improve our approach including more powerful/efficient neural operators, more efficient sparse methods, better integral approximations, and improved UNet design (Williams et al., 2023). Having demonstrated that diffusion models can be trained with $8\times$ subsampling rates, we believe there is substantial room for further performance gains. Also of interest are recent works which speed up diffusion sampling by iteratively upsampling throughout the backwards process, requiring a separate model for each resolution (Jing et al., 2022; Zhang et al., 2023); the resolution invariance of our approach permits this with a single model.

Recent diffusion advances are also complementary to our approach, these include consistency models (Song et al., 2023), stochastic interpolants (Albergo et al., 2023), Schrödinger bridges (De Bortoli et al., 2021), critically-damped diffusion (Dockhorn et al., 2022), architecture improvements (Hoogeboom et al., 2023), and faster solvers (Lu et al., 2022). Similar to our mollified diffusion, blurring has been used to improve diffusion (Rissanen et al., 2023; Hoogeboom and Salimans, 2023). Similar to GASP (Dupont et al., 2022b), other neural field GAN approaches exist such as CIPS (Anokhin et al., 2021) and Poly-INR (Singh et al., 2023), however, these approaches use convolutional discriminators requiring all coordinates on a fixed grid, preventing scaling to infinite resolutions. Also of relevance are Neural Processes (Garnelo et al., 2018; Dutordoir et al., 2022) which learn distributions over functions similar to Gaussian Processes, however, these approaches address conditional inference, whereas we construct an unconditional generative model for substantially more complex data.

Concurrent with this work, other papers independently proposed diffusion models in infinite dimensions (Lim et al., 2023; Franzese et al., 2023; Hagemann et al., 2023; Zhuang et al., 2023; Kerrigan et al., 2023; Baldassari et al., 2023; Pidstrigach et al., 2023), these approaches are complementary to ours and distinct in a number of ways. While our work focuses on the practical development necessary to efficiently model complex high-dimensional data, these papers instead focus more on theoretical foundations, typically being only applied to simple data (e.g. Gaussian mixtures and MNIST). Of particular interest, Kerrigan et al. (2023) also develop diffusion models in Hilbert space, going further than our work in foundational theory, including more on the requirements to obtain well-posed models, as well as considering different function spaces; (Lim et al., 2023) develop infinite-dimensional diffusion defined as SDEs; and Franzese et al. (2023) prove the existence of the backwards SDE. Unlike our work, these approaches make use of conditional neural fields or operate on uniform grids of coordinates, whereas our approach operates on raw sparse data, enabling better scaling. The closest to this work in terms of scaling is Diffusion Probabilistic Fields (Zhuang et al., 2023) which denoises coordinates independently using small coordinate subsets for context; this is much more restrictive than our approach and resolutions are much smaller than ours (up to $64\times64$).

## 7 CONCLUSION

In conclusion, we found that our infinite-dimensional Hilbert space diffusion model with transition densities represented by non-local integral operators is able to generate high-quality arbitrary resolution samples. Despite only observing small subsets of pixels during training, sample quality significantly surpasses prior infinite-dimensional generative models, and is competitive with state-of-the-art finite-dimensional models trained on all pixels at once. While prior infinite-dimensional approaches use latent conditional neural fields, our findings demonstrate that sparse neural operators which operate directly on raw data are a capable alternative, offering significant advantages by removing the point-wise constraint and thus the need for latent compression. Future work would benefit from improved neural operators that can effectively operate at greater levels of sparsity to improve the efficiency of our approach and enable even further scaling.

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

## A    IMPLEMENTATION DETAILS

All $256 \times 256$ models are trained on a single NVIDIA A100 80GB GPU using automatic mixed precision. Optimisation is performed using the Adam optimiser (Kingma and Ba, 2015) with a batch size of 32 and learning rate of $5 \times 10^{-5}$; each model being trained to optimise validation loss. Each model is trained as a diffusion autoencoder to reduce training variance, allowing much smaller batch sizes thereby permitting training on a single GPU. A latent size of 1024 is used and the latent model architecture and diffusion hyperparameters are the same as used by Preechakul et al. (2022). In image space, the diffusion model uses a cosine noise schedule (Nichol and Dhariwal, 2021) with 1000 steps. Mollifying is performed with Gaussian blur with a variance of 1.0.

For the image-space architecture, 3 sparse residual convolution operator blocks are used on the sparse data. Each of these consist of a single depthwise sparse convolution layer with kernel size 7 and 64 channels with the output normalised by the total number of coordinates in each local region, followed by a three layer MLP; modulated layer normalisation (Ba et al., 2016; Nichol and Dhariwal, 2021) is used to normalise and condition on the diffusion time step. These blocks use larger convolution kernels than typically used in diffusion model architectures to increase the number of coordinates present in the kernel when a small number of coordinates are sampled. Using large kernel sizes paired with MLPs has found success in recent classification models such as ConvNeXt (Liu et al., 2022).

As mentioned in Section 4.2, for the grid-based component of our architecture we experimented with a variety of U-Net shaped fourier neural operator (Li et al., 2021; Rahman et al., 2022) architectures; although these more naturally allow resolution changes at those levels, we found this came with a substantial drop in performance. This was even the case when operating at different resolutions. As such, we use the architecture used by Nichol and Dhariwal (2021) fixed at a resolution of $128 \times 128$; the highest resolution uses 128 channels which is doubled at each successive resolution up to a maximum factor of 8; attention is applied at resolutions 16 and 8, as is dropout, as suggested by Hoogeboom et al. (2023). Although this places more emphasis on the sparse operators for changes in sampling resolution, we found this approach to yield better sample quality across the board.

### A.1    TABLE 1 EXPERIMENT DETAILS

In Table 1 we compare against a number of other approaches in terms of $\text{FID}_{\text{CLIP}}$. In each case scores are calculated by comparing samples against full datasets, with 30k samples used for CelebA-HQ and 50k samples used for FFHQ and LSUN Church. For the finite-dimensional approaches we use the official code and released models in all cases to generate samples for calculating $\text{FID}_{\text{CLIP}}$ scores. To provide additional context for CelebA-HQ $128 \times 128$ we provide scores for StyleGAN2 and StyleSwin where samples are downsampled from $256 \times 256$.

For GASP (Dupont et al., 2022b) we use the official code and pretrained models for CelebA-HQ at resolutions $64 \times 64$ and $128 \times 128$; for FFHQ and LSUN Church we train models using the same generator hyperparameters as used for all experiments in that work and for the discriminator we scale the model used for CelebA-HQ $128 \times 128$ by adding an additional block to account for the resolution change. These models were trained by randomly sampling $1/4$ of coordinates at each training step to match what was used to train our models.

Similarly, for GEM (Du et al., 2021) we use the official code and train models on CelebA-HQ $128 \times 128$, FFHQ and LSUN Church, using the same hyperparameters as used for all experiments in that work (which are independent of modality/resolution); we implement sampling as described in Section 4.4 of that paper by randomly sampling a latent corresponding to a data point, linearly interpolating to a neighbouring latent and adding a small amount of Gaussian noise (chosen with standard deviation between 0 and 1 to optimise sample quality/diversity) and then projected onto the local manifold. Because GEM is based on the autodecoder framework where latents are allocated per datapoint and jointly optimised with the model parameters, training times are proportional to the number of points in the dataset. As such, the FID scores for LSUN Church in particular are especially poor due to the large scale of the dataset.

## B  MOLLIFIED DIFFUSION DERIVATION

In this section we derive the mollified diffusion process proposed in Section 3.2.

### B.1  FORWARD PROCESS

We start by specifically choosing $q(x_0|x_t)$ as defined by Section 2.1 where states are mollified by $T$,

$$q(x_{t-1}|x_0) = \mathcal{N}(x_{t-1}; \sqrt{\bar{\alpha}_{t-1}}Tx_0, (1 - \bar{\alpha}_{t-1})TT^*), \tag{17}$$

$$q(x_t|x_0) = \mathcal{N}(x_t; \sqrt{\bar{\alpha}_t}Tx_0, (1 - \bar{\alpha}_t)TT^*), \tag{18}$$

from which we wish to find the corresponding representation of $q(x_{t-1}|x_t, x_0)$. The solution to this is given by Bishop and Nasrabadi (2006, Equations 2.113 to 2.117), where we write

$$q(x_t|x_0) = \mathcal{N}(x_t; \mu, \Lambda^{-1}), \tag{19}$$

$$q(x_{t-1}|x_0) = \mathcal{N}(x_{t-1}; A\mu + b, L^{-1} + A\Lambda^{-1}A^*), \tag{20}$$

$$q(x_{t-1}|x_t, x_0) = \mathcal{N}(x_{t-1}; Ax_t + b, L^{-1}). \tag{21}$$

From this we can immediately see that $\mu = \sqrt{\bar{\alpha}_t}Tx_0$ and $\Lambda^{-1} = (1 - \bar{\alpha}_t)TT^*$. Additionally, $A\mu + b = A\sqrt{\bar{\alpha}_t}Tx_0 + b = \sqrt{\bar{\alpha}_{t-1}}Tx_0$, therefore we can modify the approach by Ho et al. (2020), including $T$ where relevant and set $A$, $b$ and $L^{-1}$ as

$$A = \frac{\sqrt{\alpha_t}(1 - \bar{\alpha}_{t-1})}{1 - \bar{\alpha}_t}I, \qquad b = \frac{\sqrt{\bar{\alpha}_{t-1}}\beta_t}{1 - \bar{\alpha}_t}Tx_0, \qquad L^{-1} = \frac{1 - \bar{\alpha}_{t-1}}{1 - \bar{\alpha}_t}\beta_t TT^* \tag{22}$$

This can be shown to be correct by passing $A$, $b$, and $L^{-1}$ into equations Eqs. (19) to (21), first:

$$A\mu + b = \frac{\sqrt{\bar{\alpha}_t}\sqrt{\alpha_t}(1 - \bar{\alpha}_{t-1})}{1 - \bar{\alpha}_t}Tx_0 + b \tag{23}$$

$$= \frac{\sqrt{\bar{\alpha}_t}\sqrt{\alpha_t}(1 - \bar{\alpha}_{t-1}) + \sqrt{\bar{\alpha}_{t-1}}(1 - \alpha_t)}{1 - \bar{\alpha}_t}Tx_0 \tag{24}$$

$$= \frac{\sqrt{\bar{\alpha}_{t-1}}(\alpha_t - \bar{\alpha}_t + 1 - \alpha_t)}{1 - \bar{\alpha}_t}Tx_0 = \sqrt{\bar{\alpha}_{t-1}}Tx_0, \tag{25}$$

and secondly,

$$L^{-1} + A\Lambda^{-1}A^* = \frac{(1 - \bar{\alpha}_{t-1})(1 - \alpha_t)}{1 - \bar{\alpha}_t}TT^* + \frac{\alpha_t(1 - \bar{\alpha}_{t-1})^2(1 - \bar{\alpha}_t)}{(1 - \bar{\alpha}_t)^2}TT^* \tag{26}$$

$$= \frac{1 - \bar{\alpha}_{t-1} - \alpha_t + \bar{\alpha}_t + \alpha_t - 2\bar{\alpha}_t + \alpha_t\bar{\alpha}_{t-1}^2}{1 - \bar{\alpha}_t} = (1 - \bar{\alpha}_{t-1})TT^* \tag{27}$$

$$= \frac{(1 - \bar{\alpha}_t)(1 - \bar{\alpha}_{t-1})}{1 - \bar{\alpha}_t}TT^* = (1 - \bar{\alpha}_{t-1})TT^*, \tag{28}$$

which together form $q(x_{t-1}|x_0)$.

## B.2 REVERSE PROCESS

In this section we derive the loss for optimising the proposed mollified diffusion process and discuss the options for parameterising the model. As stated in Section 3.2, we define the reverse transition densities as

$$p_\theta(x_{t-1}|x_t) = \mathcal{N}(x_{t-1}; \mu_\theta(x_t, t), \sigma_t^2 TT^*), \tag{29}$$

where $\sigma_t^2 = \beta_t$ or $\sigma_t^2 = \tilde{\beta}_t$. From this we can calculate the loss at time $t-1$ in the same manner as the finite-dimensional case (Eq. 3) by calculating the Kullback-Leibler divergence in infinite dimensions (Pinski et al., 2015), for conciseness we ignore additive constants throughout since they do not affect optimisation,

$$\mathcal{L}_{t-1} = \mathbb{E}_q \left[ \frac{1}{2\sigma_t^2} \left\| T^{-1}(\tilde{\mu}_t(x_t, x_0) - \mu_\theta(x_t, t)) \right\|_{\mathcal{H}}^2 \right]. \tag{30}$$

To find a good representation for $\mu_\theta$ we expand out $\tilde{\mu}_t$ as defined in Eq. (10) in the above loss giving

$$\mathcal{L}_{t-1} = \mathbb{E}_q \left[ \frac{1}{2\sigma_t^2} \left\| T^{-1} \left( \frac{\sqrt{\bar{\alpha}_{t-1}}\beta_t}{1-\bar{\alpha}_t} Tx_0 + \frac{\sqrt{\alpha_t}(1-\bar{\alpha}_{t-1})}{1-\bar{\alpha}_t} x_t - \mu_\theta(x_t, t) \right) \right\|_{\mathcal{H}}^2 \right]. \tag{31}$$

From this we can see that one possible parameterisation is to directly predict $x_0$, that is,

$$\mu_\theta(x_t, t) = \frac{\sqrt{\bar{\alpha}_{t-1}}\beta_t}{1-\bar{\alpha}_t} Tf_\theta(x_t, t) + \frac{\sqrt{\alpha_t}(1-\bar{\alpha}_{t-1})}{1-\bar{\alpha}_t} x_t. \tag{32}$$

This parameterisation is interesting because when sampling, we can use the output of $f_\theta$ to directly obtain an estimate of the unmollified data. Additionally, when calculating the loss $\mathcal{L}_{t-1}$, all $T$ and $T^{-1}$ terms cancel out meaning there are no concerns with reversing the mollification during training, which can be numerically unstable. To see this, we can further expand out Eq. (31) using the parameterisation of $\mu_\theta$ defined in Eq. (32),

$$\mathcal{L}_{t-1} = \mathbb{E}_q \left[ \frac{1}{2\sigma_t^2} \left\| T^{-1} \left( \frac{\sqrt{\bar{\alpha}_{t-1}}\beta_t}{1-\bar{\alpha}_t} Tx_0 + \frac{\sqrt{\alpha_t}(1-\bar{\alpha}_{t-1})}{1-\bar{\alpha}_t} x_t \right. \right. \right.$$
$$\left. \left. \left. - \frac{\sqrt{\bar{\alpha}_{t-1}}\beta_t}{1-\bar{\alpha}_t} Tf_\theta(x_t, t) - \frac{\sqrt{\alpha_t}(1-\bar{\alpha}_{t-1})}{1-\bar{\alpha}_t} x_t \right) \right\|_{\mathcal{H}}^2 \right] \tag{33}$$

$$= \mathbb{E}_q \left[ \frac{1}{2\sigma_t^2} \left\| \frac{\sqrt{\bar{\alpha}_{t-1}}\beta_t}{1-\bar{\alpha}_t} x_0 - \frac{\sqrt{\bar{\alpha}_{t-1}}\beta_t}{1-\bar{\alpha}_t} f_\theta(x_t, t) \right\|_{\mathcal{H}}^2 \right] \tag{34}$$

$$= \mathbb{E}_q \left[ \frac{\sqrt{\bar{\alpha}_{t-1}}\beta_t}{2\sigma_t^2(1-\bar{\alpha}_t)} \left\| x_0 - f_\theta(x_t, t) \right\|_{\mathcal{H}}^2 \right]. \tag{35}$$

Using this parameterisation, we can sample from $p_\theta(x_{t-1}|x_t)$ as

$$x_{t-1} = \frac{\sqrt{\alpha_t}(1-\bar{\alpha}_{t-1})}{1-\bar{\alpha}_t} x_t + T \left( \frac{\sqrt{\bar{\alpha}_{t-1}}\beta_t}{1-\bar{\alpha}_t} f_\theta(x_t, t) + \sigma_t \xi \right) \quad \text{where} \quad \xi \sim \mathcal{N}(0, C_I). \tag{36}$$

Alternatively, we can parameterise $\tilde{\mu}$ to predict the noise $\xi$ rather than $x_0$, which was found by Ho et al. (2020) to yield higher sample quality. To see this, we can write Eq. (9) as $x_t(x_0, \xi) = \sqrt{\bar{\alpha}_t} Tx_0 + \sqrt{1-\bar{\alpha}_t} T\xi$. Expanding out Eq. (30) with this gives the following loss,

$$\mathcal{L}_{t-1} = \mathbb{E}_q \left[ \frac{1}{2\sigma_t^2} \left\| T^{-1} \left( \tilde{\mu}(x_t, x_0) - \mu_\theta(x_t, t) \right) \right\|_{\mathcal{H}}^2 \right] \tag{37}$$

$$= \mathbb{E}_q \left[ \frac{1}{2\sigma_t^2} \left\| T^{-1} \left( \tilde{\mu}(x_t(x_0, \xi), \frac{1}{\sqrt{\bar{\alpha}_t}} T^{-1}(x_t(x_0, \xi) - \sqrt{1-\bar{\alpha}_t} T\xi)) - \mu_\theta(x_t, t) \right) \right\|_{\mathcal{H}}^2 \right] \tag{38}$$

$$= \mathbb{E}_q \left[ \frac{1}{2\sigma_t^2} \left\| T^{-1} \left( \frac{1}{\sqrt{\alpha_t}} \left( x_t(x_0, \xi) - \frac{\beta_t}{\sqrt{1-\bar{\alpha}_t}} T\xi \right) - \mu_\theta(x_t, t) \right) \right\|_{\mathcal{H}}^2 \right]. \tag{39}$$

Since directly predicting $\xi$ would require predicting a non-continuous function, we instead propose predicting $T\xi$ which is a continuous function, giving the following parameterisation and loss,

$$\mu_\theta(x_t, t) = \frac{1}{\sqrt{\alpha_t}} \left[ x_t - \frac{\beta_t}{\sqrt{1 - \bar{\alpha}_t}} f_\theta(x_t, t) \right], \tag{40}$$

$$\mathcal{L}_{t-1} = \mathbb{E}_q \left[ \frac{1}{2\sigma_t^2} \left\| \frac{1}{\sqrt{\alpha_t}} T^{-1} \left( \frac{\beta_t}{\sqrt{1 - \bar{\alpha}_t}} f_\theta(x_t, t) - \frac{\beta_t}{\sqrt{1 - \bar{\alpha}_t}} T\xi \right) \right\|_{\mathcal{H}}^2 \right]. \tag{41}$$

In this case $T^{-1}$ is a linear transformation that does not affect the minima. In addition to this, we can remove the weights as suggested by Ho et al. (2020), giving the following proxy loss,

$$\mathcal{L}_{t-1}^{\text{simple}} = \mathbb{E}_q \left[ \| f_\theta(x_t, t) - T\xi \|_{\mathcal{H}}^2 \right]. \tag{42}$$

An alternative parameterisation which can train more reliably is $v$-prediction (Salimans and Ho, 2022; Hoogeboom et al., 2023); we experimented with this parameterisation but found mollified noise prediction to yield higher quality samples.

## C   ADDITIONAL RESULTS

In this section, we provide additional details on the training speedup and memory reductions possible using our multi-scale architecture, from sparse coordinate sampling at different rates, and with different fixed resolutions of the inner U-Net. In Table 4 we calculate the training speedup for a fixed memory budget of 10GB, while in Table 5 we calculate the memory reduction for a fixed batch size of 16. Additionally, we add here an extra quantitative result to assess sample quality at when sampling from the FFHQ model trained only on $256 \times 256$ images at a resolution of $1024 \times 1024$: calculating $\text{FID}_{\text{CLIP}}$ on 5000 samples using 50 sampling steps, the model achieves a score of 15.84. We also provide additional samples from our models to visually assess quality. Detecting overfitting is crucial when training generative models. Scores such as FID are unable to detect overfitting, making identifying overfitting difficult in approaches such as GANs. Because diffusion models are trained to optimise a bound on the likelihood, training can be stopped to minimise validation loss. As further evidence we provide nearest neighbour images from the training data to samples from our model, measured using LPIPS (Zhang et al., 2018).

| | Data Resolution | 128 | | 256 | | 512 | |
|---|---|---|---|---|---|---|---|
| | Inner Resolution | 32 | 64 | 64 | 128 | 128 | 256 |
| Rate | 2× | 1.00× | 1.00× | 1.00× | 1.00× | 1.00× | ∞× |
| | 4× | 1.64× | 1.53× | 1.97× | 2.08× | 3.61× | ∞× |
| | 8× | 2.65× | 2.17× | 3.51× | 3.18× | 7.98× | ∞× |
| | 16× | 2.89× | 2.49× | 4.57× | 3.83× | 11.78× | ∞× |

Table 4: Training speedup for a fixed memory budget. Larger is better. Calculated for different subsampling rates, training data resolutions, and resolutions of the grid within the multi-scale architecture.

| | Data Resolution | 128 | | 256 | | 512 | |
|---|---|---|---|---|---|---|---|
| | Inner Resolution | 32 | 64 | 64 | 128 | 128 | 256 |
| Rate | 2× | 1.56× | 1.44× | 1.42× | 1.52× | 1.43× | 1.32× |
| | 4× | 2.44× | 2.02× | 1.99× | 1.66× | 2.00× | 1.67× |
| | 8× | 3.27× | 2.46× | 2.44× | 1.88× | 2.44× | 1.90× |
| | 16× | 3.90× | 2.75× | 2.73× | 2.01× | 2.73× | 2.03× |

Table 5: Memory reduction during training for a fixed batch size. Larger is better. Calculated for different subsampling rates, training data resolutions, and resolutions of the grid within the multi-scale architecture.

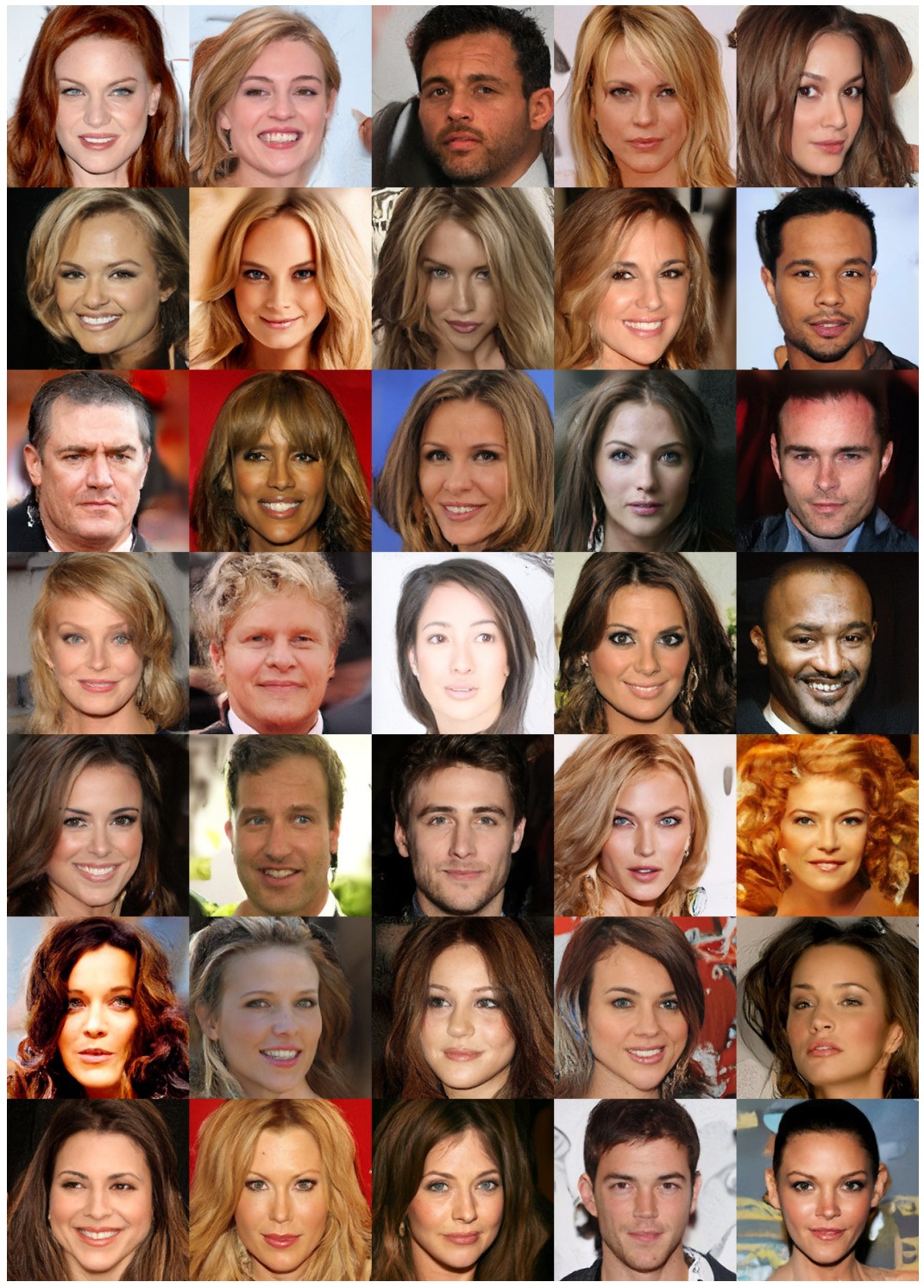

Figure 10: Non-cherry picked, CelebA-HQ $256 \times 256$ samples.

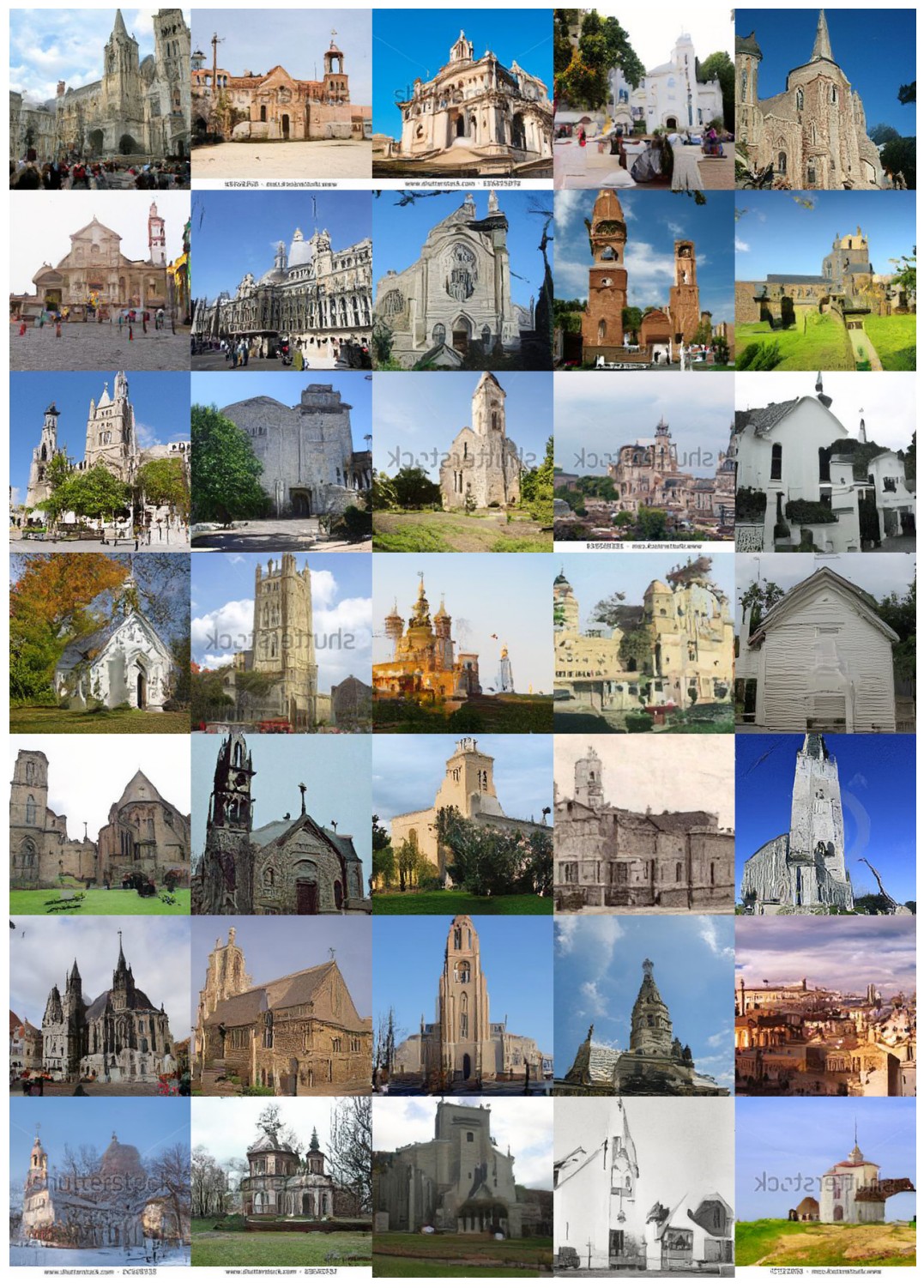

Figure 11: Non-cherry picked, LSUN Church $256 \times 256$ samples.

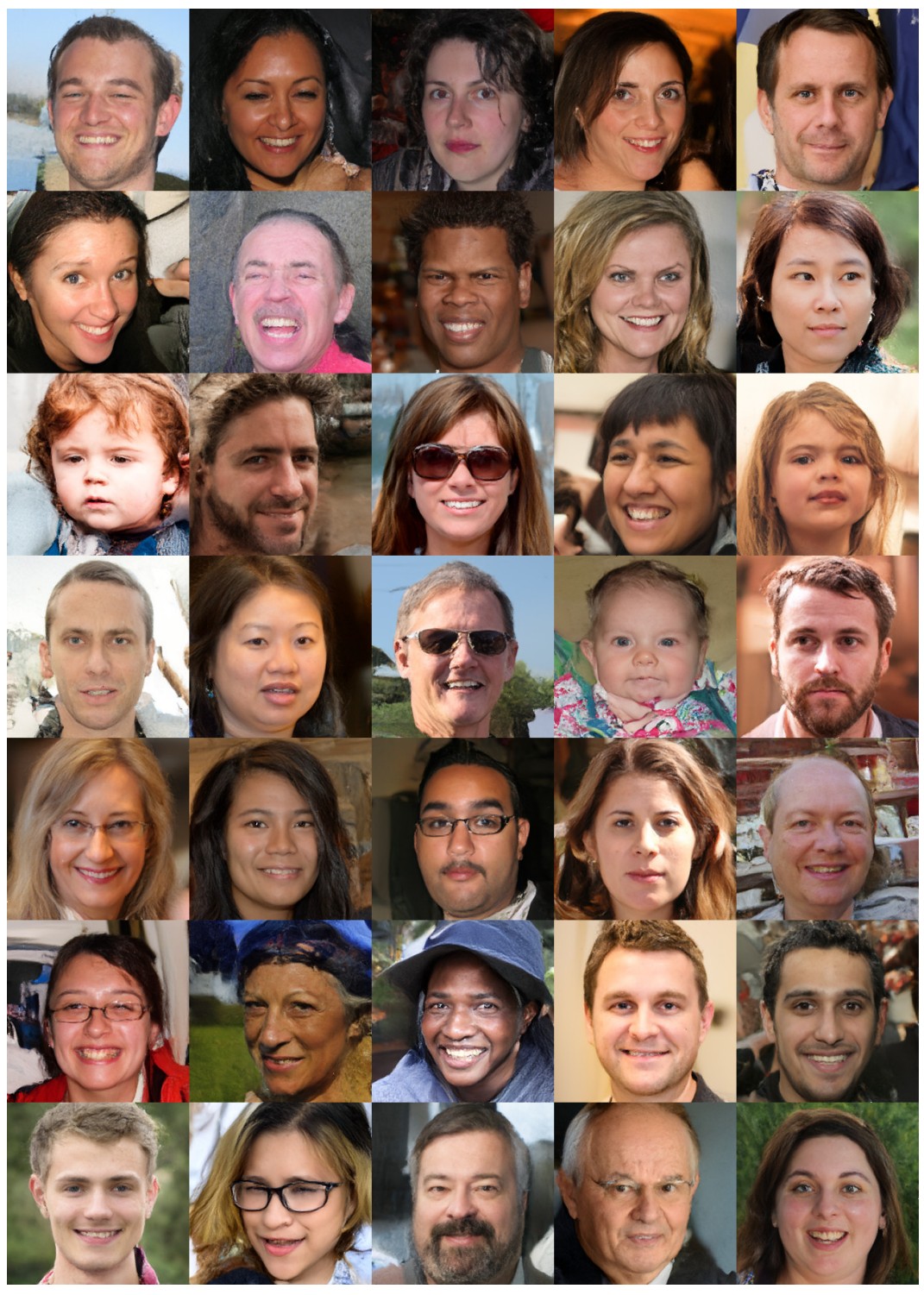

Figure 12: Non-cherry picked, FFHQ $256 \times 256$ samples.

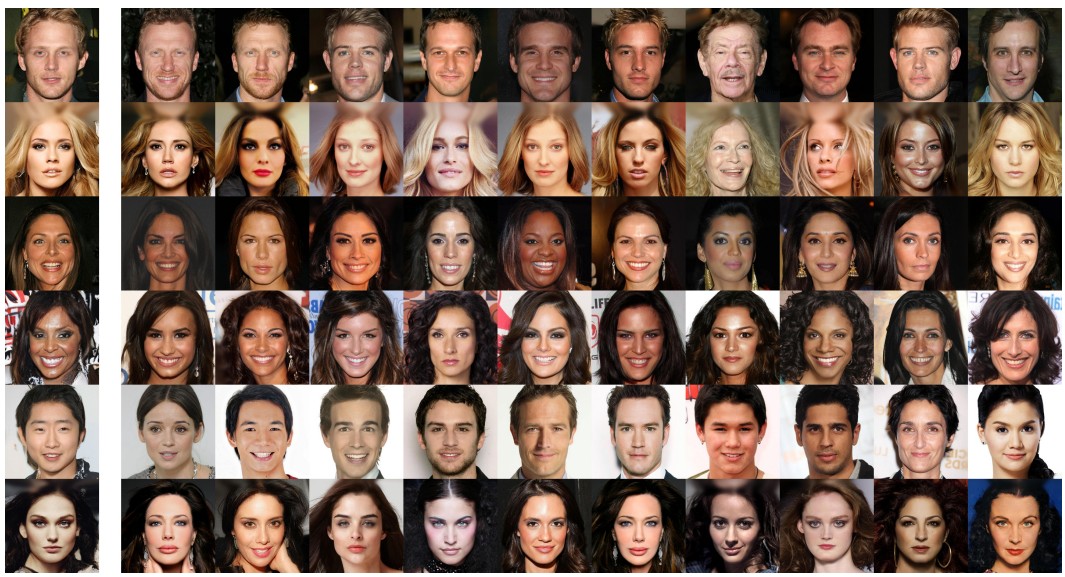

Figure 13: Nearest neighbours for a model trained on CelebA-HQ based on LPIPS distance. The left column contains samples from our model and the right column contains the nearest neighbours in the training set (increasing in distance from left to right)

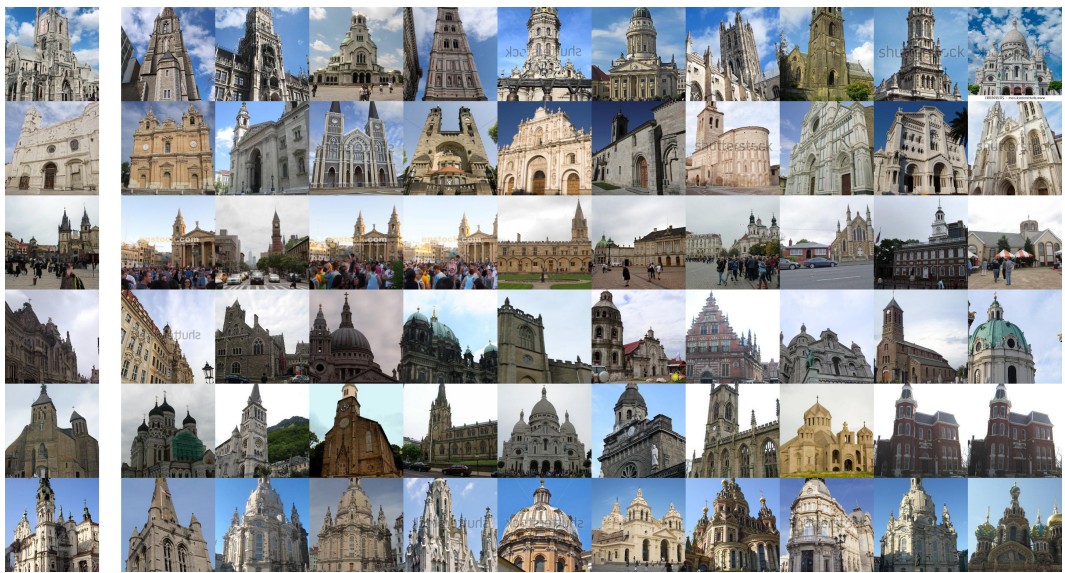

Figure 14: Nearest neighbours for a model trained on LSUN Church based on LPIPS distance. The left column contains samples from our model and the right column contains the nearest neighbours in the training set (increasing in distance from left to right)

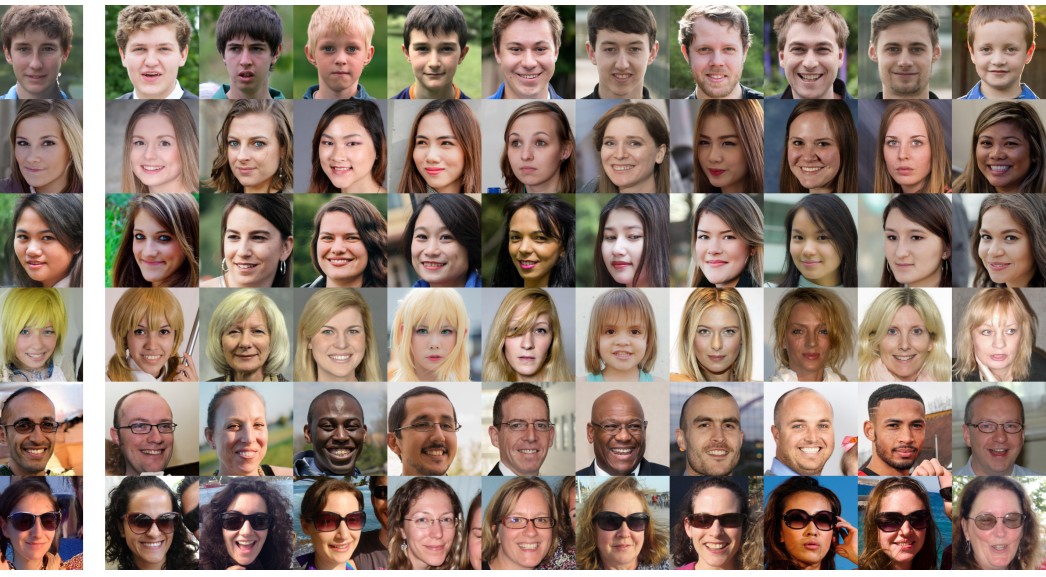

Figure 15: Nearest neighbours for a model trained on FFHQ based on LPIPS distance. The left column contains samples from our model and the right column contains the nearest neighbours in the training set (increasing in distance from left to right)

