# OpenReview forum: "$\infty$-Diff: Infinite Resolution Diffusion with Subsampled Mollified States"
_ICLR.cc/2024/Conference — ICLR 2024 poster_

### Official Review · Reviewer_btvx · 2023-10-30

**Soundness:** 2 fair
**Presentation:** 2 fair
**Contribution:** 3 good
**Rating:** 6
**Confidence:** 4

**Summary:**

**Edit: I have raised my score to 6 to reflect the author's updates to the submission.**

This paper proposes a generalization of diffusion models for data living in infinite dimensional Hilbert spaces. The main motivation for doing so is to enable models which can be trained and sampled at arbitrary resolutions and to enable scaling of diffusion models to high-dimensional data. Section 3 discusses some theoretical concerns for developing such models, Section 4 proposes an architecture well-suited for this task on natural image data, and Section 5 provides an empirical evaluation of the proposed methodology.

**Strengths:**

- The proposed neural architecture is novel and a significant contribution to the literature on infinite-dimensional models. In particular, the proposed model shows significant performance gains in comparison to several previously proposed infinite-dimensional generative models (Table 1) in terms of FID scores.
- The experiments are generally well-executed and convincing in terms of the conclusions drawn from these throughout the paper.

**Weaknesses:**

- A major weakness of this paper is how the authors frame their contributions within the existing literature.
     - For example, in the abstract the authors write "Unlike prior infinite-dimensional models, which use point-wise functions requiring latent compression, our method employs non-local integral operators to map between Hilbert spaces, allowing spatial context aggregation." Similarly, in Section 1, the authors write "We introduce a new Gaussian diffusion model defined in an infinite-dimensional state space that allows infinite resolution data to be generated (see Fig. 2)". However, numerous recent works have posed infinite-dimensional diffusion models which precisely use these integral operators and a similar theoretical framework [1, 2, 3, 4, 5, 6, 7].
     - The authors are clearly aware of this work (see Section 6) but do not appropriately frame their contributions, i.e. many of these prior works develop the theory of infinite-dimensional models which is very closely related to the proposed theory, but the authors of this submission do not state this. The authors claim that these works are concurrent, but the earliest of these works appeared in December 2022 [4], February 2023 [1], and March 2023 [2, 3], more than 6 months prior to the ICLR submission deadline. In addition, the authors are missing a citation to Kerrigan et al. [4] who previously develop a theory for discrete-time diffusion models which is closely related to the theory the authors propose in Section 3, as well as references to several other infinite-dimensional diffusion models in continuous time [5, 6, 7].

- The theory in Section 3 is imprecise to the degree of incorrectness.
     - The authors write "The Radon-Nikodym theorem tells us the density for a measure $\nu$ absolutely continuous with respect to a base measure $\mu$". This is not what the Radon-Nikodym says -- the Radon-Nikodym theorem states the *existence* of a density given the absolute continuity. To actually compute this density, you require stronger results, such as the Cameron-Martin and Feldman-Hajek theorems. See [1, 4] for a discussion of these theorems in the context of diffusion models.
     - The authors miss key regularity assumptions necessary on the Gaussian noise in order to obtain a well-posed model, see again [1, 4] for a discussion of these requirements.


### Minor
- The discussion of the "diffusion autoencoder framework" in Section 5 could use a short description for those unfamiliar with the work, i.e. it was not entirely clear to me how model is trained based on the description provided alone.


### References:
[1] [Lim et al., Score-Based Diffusion Models in Function Space](https://arxiv.org/abs/2302.07400)

[2] [Franzese et al., Continuous-Time Functional Diffusion Processes](https://arxiv.org/abs/2303.00800)

[3] [Hagemann et al., Multilevel Diffusion: Infinite Dimensional Score-Based Diffusion Models for Image Generation](https://arxiv.org/abs/2303.04772)

[4] [Kerrigan et al., Diffusion Generative Models in Infinite Dimensions](https://arxiv.org/abs/2212.00886)

[5] [Lim et al., Score-based Generative Modeling through Stochastic Evolution Equations
](https://neurips.cc/virtual/2023/poster/72191)

[6] [Baldassari et al., Conditional score-based diffusion models for Bayesian inference in infinite dimensions](https://arxiv.org/abs/2305.19147)

[7] [Pidstrigach et al., Infinite-Dimensional Diffusion Models](https://arxiv.org/abs/2302.10130)

**Questions:**

See weakness section.

Overall, I think this work provides an important and novel practical contribution (namely an empirically sound architecture for infinite-dimensional diffusions), but the contributions regarding the theory are over-stated and the framing of the work with regards to prior work in this area needs to be significantly improved.

---

> ### Author Response · Authors · 2023-11-20
> **Response to Reviewer btvx**
>
> > **Numerous recent works have posed infinite-dimensional diffusion models… [The authors] do not appropriately frame their contributions**
>
> The ICLR guidelines state that these works are considered contemporaneous, and we are not required to compare our paper to them. Specifically, it states that if a paper has not been published at a peer-reviewed venue by May 28th then it is considered contemporaneous. The papers we discuss as concurrent work either exist only on arXiv, or were published after this date (e.g. at NeurIPS 2023). Additionally, our paper was published on arXiv during the months the reviewer states as the “earliest of these works”, in fact, some of these works cite our paper as concurrent work. As such, we believe it is fair to describe our paper as a concurrent work.
>
> > **“The authors are missing a citation to Kerrigan et al. [4]” and “The contributions regarding the theory are over-stated”**
>
> Thanks for pointing out this missing citation; we were unaware of it and did not intend to overstate our contributions. While our approach was developed entirely independently, with our paper published on arXiv early this year before the conference proceedings of this work, we believe it is important to properly credit prior work. As such, in the introduction, we mention that concurrent works also developed diffusion models in infinite-dimensions; at the start of Section 3 we state that these concurrent works exist, and that they delve deeper into the theory than our paper; and finally, we have expanded the discussion on the concurrent works in Section 6. Throughout the paper we have reduced the strength of our contribution claims and/or also mentioned concurrent work (including those pointed out by the reviewer), and in parts where details are limited compared to these concurrent works (such as on the mentioned Radon-Nikodym theorem and regularity assumptions) we have added references to appropriate papers and stated their contributions.
>
> Nonetheless, in terms of the practical aspect, we believe our approach offers substantial contributions extending these works since it is a major challenge to efficiently scale such models to enable practical modelling of complex data. Kerrigan et al. evaluate their approach only on small 2D curve datasets, while the other concurrent works evaluate on simple datasets such as Gaussian mixtures or MNIST (with FID scores substantially worse than baseline finite-dimensional approaches). Our work instead focuses on scaling to substantially more complex, high resolution datasets, developing a practical architecture, subsampling coordinates to achieve speedup/memory gains, and achieving state-of-the-art results for infinite-dimensional models by a wide margin, which presents a very large challenge as highlighted by the other reviewers. As such, with emphasis on such different aspects, we believe our work and each of these concurrent works are very complementary.
>
> > **Discussion on diffusion autoencoder framework**
>
> We have added more details on the diffusion autoencoder framework to this section to explain how they can be used to reduce stochasticity and allow training with smaller batch sizes. Specifically, that an encoding network is introduced (which in our case operates on sparsely sampled coordinates) to map data points to small latent vectors; our mollified diffusion process is then conditioned on these latent vectors meaning that at the high time steps when the input to the denoising network is mostly noise, the latent provides additional information that allows better estimation of the denoised data, reducing gradient stochasticity. Subsequently, a small diffusion model with MLP backbone can be quickly trained to model these latents to allow unconditional sampling.

---

> > ### Comment · Reviewer_btvx · 2023-11-20
> >
> > I thank the authors for their detailed response. I would like to clarify that I do not expect the authors to perform an experimental comparison to the methods listed, given their (relative) contemporaneity. I believe the changes made to the submission regarding the existing literature in this area is sufficient.
> >
> > I am willing to raise my score to a 6 given these changes to the submission. However, there still remain a number of errors with regards to the theoretical claims in Section 3. While this paper is primarily a practical contribution, these errors would ideally be fixed in a later version of the paper.

---

### Official Review · Reviewer_iaub · 2023-10-31

**Soundness:** 4 excellent
**Presentation:** 4 excellent
**Contribution:** 4 excellent
**Rating:** 8
**Confidence:** 5

**Summary:**

The paper considers the generative modeling problem and the diffusion models in particular. The proposed novel diffusion model is defined in an infinite-dimensional Hilbert space in order to possibly model infinite resolution data. The model is trained only on randomly sampled subsets of coordinates and denoising data only there. It allows for learning a continuous function for arbitrary resolution sampling. Whereas the standard infinite-dimensional models use point-wise functions requiring latent compression, the proposed model employs non-local integral operators to map between Hilbert spaces and, as such, allows spatial context aggregation. The method is compared against the state-of-the-art models across different tasks achieving comparable results.

**Strengths:**

The paper has a few significant strengths overall, which I will outline below:
1. Proposed model achieve the best or comparable results to the state-of-the-art models.
2. Formulating the generative diffusion model in an infinite-dimensional Hilbert space and allowing to denoise data only on a subset of coordinates is very interesting. I would assume that it might be to complex problem but maybe the mollification is why the model is able learn?
3. Formulating the diffusion model in a Hilbert space in a strict mathematical regime, which seems to be correct.
4. Including additional experiments on not so common tasks like inpainting.
5. The presentation is very clear. Overall, the flow of the manuscript is well-organized.

**Weaknesses:**

However, despite the strengths, the paper has a few major and minor weaknesses:
1. The works based on similar ideas are mentioned only in the Discussion section, but the proposed model is not compare against them in an experimental way.
2. The lack of information regarding the computational costs of both training and inference for the proposed model.
3. I couldn’t find the information which is actual upscaling procedure limit, which will be significant from the practitioner point of view.

**Questions:**

I would like to see especially the following experiments and improvements regarding specifically to the Weaknesses section:
1. It will be great if the authors might compare the works mentioned in the Discussion section against their model in the same experimental settings.
2. I would like to see a new experiment comparing the computational costs of proposed model on different resolutions (e.g., in FLOPs).
3. The authors show the results on up to 8x subsampling rate but I’m considering what will be the highest subsampling rate (and resolution) when the model is still doing well. Please, if you could include such comparison.

---

> ### Author Response · Authors · 2023-11-20
> **Response to Reviewer iaub**
>
> We thank the reviewer for their very helpful feedback and positive comments.
>
> > **The proposed model is not compared against [the similar works] in an experimental way**
>
> Our approach is designed to address the problem of efficient training on high resolution datasets, which we achieve through the combination of our mollified diffusion framework and efficient sparse architecture design. In contrast, the architectures alone used in these concurrent works do not scale well, making training on these impractical (in terms of memory and compute), where these works only consider very simple toy datasets such as Gaussian mixtures or MNIST with FID scores substantially worse than baseline finite-dimensional approaches. These works are considered contemporaneous, which we are not required to compare against.
>
> > **Lack of information regarding the computational costs of both training and inference**
>
> We agree that more information on computational costs, beyond the details in Section 5 and Table 3, would be useful. Particularly when applied to different resolutions, to assess how our approach scales. Sampling enough images for FID calculations is particularly slow at the highest resolutions, so will add more experiments into the final manuscript. Specifically, we will make a table with various combinations (e.g. data resolution, number of sampled coordinates, and inner grid resolution) and show run-time and memory usage.
>
> > **[What is the] actual upscaling procedure limit?**
>
> Theoretically, the potential maximum resolution is infinite (technically bounded only by computation constraints and floating point precision), since the nature of the model is continuous. While at higher resolutions than the training data, you can’t expect results that contain substantially greater detail than the training data, we do find in Figure 7 sampling at higher resolutions does achieve good FID scores. However, as observed in Figure 2, we find that samples converge in terms of details and as observed by reviewer RCWm, can contain artefacts. Although this is, to the best of our knowledge, the first class of infinite-dimensional diffusion model that can capture non-blurry results higher than the training distribution. In the final manuscript we will add FID scores for even higher resolutions and observe how FID is affected.

---

> > ### Comment · Reviewer_iaub · 2023-11-20
> >
> > I want to thank the authors for their response to my concerns and suggestions. I will be looking forward to see the additional experimental details (like FID scores, FLOPs, etc.) that have been promised by the authors. From my perspective, the most important question related to the paper is the actual scaling limits of the proposed method. I agree that we don’t have any theoretical constraints, but as always, there might be some limits (due to variety of biases) that prevent the method from scaling over the specific threshold. It will be great to see the extended discussion on this topic in the revised or final version of the manuscript.

---

### Official Review · Reviewer_RCWm · 2023-11-01

**Soundness:** 3 good
**Presentation:** 1 poor
**Contribution:** 3 good
**Rating:** 8
**Confidence:** 3

**Summary:**

This paper introduces an infinite-dimensional diffusion model to handle images at arbitrary resolutions. The authors introduce an architecture based on neural operators that achieves state-of-the-art FID among infinite-dimensional approaches at resolutions up to 256x256.

**Strengths:**

The network architecture seems to have been well designed, and the results are very convincing. In particular, the use of irregular grids seems to be a good tradeoff between efficiency and retaining fine detail information (particularly if the sampling grid varies from training point to training point, which is not clear).

**Weaknesses:**

- The paper lists the extension of diffusion models to infinite-dimensional spaces as one of its contributions, but there are already many works on this topic [1-5] (of which only [2-4] are briefly mentioned in the paper). The authors should discuss the relationship of their framework to this related work in greater detail (and they should be mentioned before the discussion). Also, the mathematical treatment of the extension to infinite dimensions in the paper is lacking compared to [1-5] and thus cannot be considered a contribution.
- While the paper aims to work at any resolution, the use of "mollification" (I suggest to simply call this blurring which is a better-known term in the NeurIPS community) on the data distribution effectively limits the maximum resolution of the generated images. A close inspection of Figure 2 also reveals artifacts on the images generated at resolution larger than the training resolution. On a related note, the authors never discuss the mechanisms by which the score network should be expected to generate to higher resolutions while never having observed high-resolution images. Could the authors expand on this point?
- The central contribution of the paper seems to be its architecture. While its components are described rather abstractly in Section 4, I do not understand what is being implemented exactly. The discretization of the integrals is never discussed in detail, as well as the grid on which the various intermediate activations are represented. When using a regular grid, what is the difference between the proposed architecture and a regular CNN such as UNet? As another important point, how discretization invariance is obtained when using images with different resolutions is not discussed. Does the number of points used in the finite-sum approximation of the integrals changes?
- Super-resolution and inpainting are rather unfaithful to the original images: downsampling or cropping the generated image does not yield back the input image.

[1] Kerrigan, Gavin, Justin Ley, and Padhraic Smyth. “Diffusion Generative Models in Infinite Dimensions.” arXiv, February 24, 2023. https://doi.org/10.48550/arXiv.2212.00886.

[2] Lim, Jae Hyun, Nikola B. Kovachki, Ricardo Baptista, Christopher Beckham, Kamyar Azizzadenesheli, Jean Kossaifi, Vikram Voleti, et al. “Score-Based Diffusion Models in Function Space.” arXiv, February 14, 2023. https://doi.org/10.48550/arXiv.2302.07400.

[3] Hagemann, Paul, Sophie Mildenberger, Lars Ruthotto, Gabriele Steidl, and Nicole Tianjiao Yang. “Multilevel Diffusion: Infinite Dimensional Score-Based Diffusion Models for Image Generation.” arXiv, April 29, 2023. https://doi.org/10.48550/arXiv.2303.04772.

[4] Franzese, Giulio, Giulio Corallo, Simone Rossi, Markus Heinonen, Maurizio Filippone, and Pietro Michiardi. “Continuous-Time Functional Diffusion Processes.” arXiv, July 7, 2023. https://doi.org/10.48550/arXiv.2303.00800.

[5] Pidstrigach, Jakiw, Youssef Marzouk, Sebastian Reich, and Sven Wang. “Infinite-Dimensional Diffusion Models.” arXiv, October 3, 2023. http://arxiv.org/abs/2302.10130.

**Questions:**

- What is the "pointwise" constraint that is mentioned in the abstract and conclusion? Perhaps related, could the authors elaborate what is meant by "coordinates are treated independently" in neural fields?
- In section 4.1, what does the notation $c \in {m \choose D}$ means?
- In equation (13), I am confused by the use of both $x$ and $v_l$. I do not understand if the operator represents a function from $x$ to $s$ or $v_l$ to $v_{l+1}$.
- I did not understand the second paragraph of section 5 on diffusion autoencoder framework. What do the authors mean by "using the first half of our architecture"?
- What do the authors mean by "each model being trained to optimize validation loss" in Appendix A? Is validation data used to train the networks on top of training data?

---

> ### Author Response · Authors · 2023-11-20
> **Response to Reviewer RCWm**
>
> Thank you for the comprehensive feedback and noting the convincing results. The feedback is much appreciated and strengthened our paper; we’ve addressed the weaknesses and questions point-by-point below:
>
> > **Already many works on this topic [1-5]… The authors should discuss the relationship of their framework to this related work in greater detail.**
>
> In the updated paper we have expanded the discussion on these works in Section 6, expanding on each paper, including a discussion on [1], and how our paper fits in. We have also added references throughout the paper to indicate additional mathematical treatment in these works.
>
> > **Mollification… effectively limits the maximum resolution of the generated images**
>
> By mollifying, we in effect suppress high-frequency details beyond a specific scale yielding smoother functions with characteristics resembling those from a finite-dimensional subspace. However, it importantly facilitates the recovery of high-frequency details under certain conditions, an advantage over information loss induced by coarse discretisation—as showcased in our super-resolution results (Fig. 8). In practice, we only need to use a very small variance blur, which corresponds to very high resolutions, far exceeding the memory of the subsampled coordinates (up to 8x tested in Table 2), yet still lying in Hilbert space to allow for arbitrary resolution sampling (Figs. 2 and 7) and super-resolution (Fig. 8).
>
> > **Generations at higher resolutions**
>
> The field of neural operators focuses on learning discretisation invariant operators and has been shown in prior work to generalise to resolutions greater than the training data (Li et al. 2021). While there is always potential for artefacts when sampling at such higher resolutions (also observed with other infinite-dimensional models), in Fig. 7 we evaluate FID at higher resolutions and observe good scores, showing that samples are high quality.
>
> > **How the architecture is implemented**
>
> At the sparse level integrals are Monte-Carlo approximated: Eqn. 16 becomes $x(c) \approx \frac{|N(c)|}{|\mathbf{c}|}\sum_{y\in\mathbf{c}} \kappa(c-y)v(y)$. Sampled coordinates do vary across training points, and the same coordinates are used through the forward pass. We found global operators worse and be less scalable than local operators so we interpolate points to/from a regular grid where a model such as a U-NO (Rahman et al. 2022) can be used for global context.
>
> When on a regular grid, there are some key differences to regular UNets. Primarily that each layer has a different formulation than typical CNN blocks, see Equation 14. The design of these layers is crucial for generalisation; for example, in Table 2 we show that if the operator kernel is non-linear (the overall operator is still non-linear), it is unable to generalise to different numbers of coordinates/resolutions.
>
> > **Super-resolution and inpainting are unfaithful to the original images**
>
> Obtaining better faithfulness when inpainting with diffusion models is a popular research area. Diffusion only in masked areas leads to edge artefacts so approximation approaches such as DDIM inversion (Song et al. 2021) and guidance (Ho et al. 2022b; which we use) were proposed, avoiding artefacts at the expense of perfect reconstruction. More recent work substantially improving this [1] can be directly applied to our method.
>
> > **Elaborate what is meant by "coordinates are treated independently"**
>
> Neural fields use an MLP to directly map coordinates to pixel values, e.g for $c \in [0,1]^2$, the corresponding pixel is $p=\text{MLP}(c)$ (Sec 2.2). Hence they are independent (pointwise) because the mapping function does not transform over multiple points unlike convolutions/transformers. We have clarified this in Section 2.2.
>
> > **What does $c \in \left(\begin{smallmatrix}D\\\\m\end{smallmatrix}\right)$**
>
> It is a subset of length $m$ from $D$, where $D=[0,1]^2$ is the set of all coordinates. In practice we uniformly sample m coordinates, but other methods exist (Sec 4.2). We have clarified this in the updated manuscript.
>
> > **Confused between $x$ to $s$ or $v_l$ to $v_{l+1}$**
>
> We use $\mathcal{F}$ to denote the entire neural operator, mapping from $\mathcal{X}$ to $\mathcal{S}$, and use $v$ to indicate the activations of the network. Therefore $x=v_0$ and $s=v_L$. We have clarified this in this section.
>
> > **What does “first half of our architecture” mean?**
>
> Following Preechakul et al., 2022, the encoder is the downsampling stage of our architecture with the upsampling and skip connections removed: looking at Figure 4, it is only the left half of the image. We have added more details on diffusion autoencoders to Section 5.
>
> > **What does optimise validation loss mean?**
>
> We only train on the training data. We use a kept-out validation set to tune hyperparameters, in this case when to stop.
>
> [1] Huberman-Spiegelglas et al. An Edit Friendly DDPM Noise Space: Inversion and Manipulations. arXiv:2304.06140 2023.

---

> > ### Comment · Reviewer_RCWm · 2023-11-21
> >
> > I thank the authors for their detailed answer which addresses my questions. I have raised my score accordingly.

---

### Official Review · Reviewer_gZH8 · 2023-11-01

**Soundness:** 3 good
**Presentation:** 2 fair
**Contribution:** 2 fair
**Rating:** 6
**Confidence:** 3

**Summary:**

In this work, the mollified diffusion models is proposed to process infinite resolution data efficiently and effectively, to overcome the weakness of low sampling speed of existing diffusion models, where the mollification of data, Fourier neural operators, multi-scale architectures and efficient sparse operators are applied. The experiments are conducted accordingly with impressive results.

**Strengths:**

1. Originality: It is a novel idea to apply mollification for diffusion models  to enable the data to be processed in the Hilbert space and allow diffusion models to generate high resolution or even infinite resolution data with efficacy. Smoothing using mollification can ensure the regularity of data and enhance enable the modelling in the Hilbert space to reduce the sampling speed.

2. Quality: The experimental results seem to be promising and impressive quantitatively and qualitatively. FIDs of the proposed $\infty$-Diff are comparative to those of finite-dimensional methods.

3. Significance: In this paper, regularity is discussed in diffusion models. It opens the horizon of future theoretical analysis of diffusion models.

**Weaknesses:**

The writing really needs to be polished. A lot of typos should be corrected, for example, Page 4 Section 3.2 Paragraph 1 Line 11 $\mu_\theta: \mathcal{H}, \mathbb{R} \rightarrow \mathcal{H}$ → $\mu_\theta: \mathcal{H} \times \mathbb{R} \rightarrow \mathcal{H}$; Page 5 Section 4 .1  Paragraph 2 Line 1-2 “respectfully” → “respectively”; etc.

**Questions:**

Q1. What does $x_t(x_0, \psi)$ mean in Equation (11)? What is the difference between $x_t(x_0, \psi)$ and $x_t$?

Q2. Could the authors provide any theoretical justification and guarantee on why mollification is necessary and suitable for diffusion models to analyze infinite resolution data?

Q3. Would the authors clarify what $v_0$ means above Equation (14)?

---

> ### Author Response · Authors · 2023-11-20
> **Response to Reviewer gZH8**
>
> We thank the reviewer for their feedback and helpful suggestions and for noting the originality, quality, and significance of our approach. We address the weaknesses and questions below point-by-point.
>
> > **Typos should be corrected**
>
> Thanks for spotting these typos, we have corrected them and revised similar cases in the updated paper.
>
> > **What does $x_t(x_0, \xi)$ mean? What is the difference between $x_t(x_0, \xi)$ and $x_t$**
>
> $x_t(x_0, \xi)$ is a reparameterization of $x_t \sim q(x_t | x_0)$. That is, for a data point $x_0$ and Gaussian noise sample $\xi$, $x_t(x_0, \xi)=\sqrt{\bar{\alpha}_t}x_0 + \sqrt{1-\bar{\alpha}_t}\xi$. We have added this definition to the updated paper.
>
> > **Theoretical justification and guarantee on why mollification is necessary and suitable**
>
> Gaussian white noise for finite-dimensional diffusion models is defined such that each dimension is independent. In infinite-dimensional Hilbert spaces, white noise defined similarly exhibits discontinuities that mean that it does not lie in this space. Specifically, this is because it does not satisfy the trace-class requirement, that is, that the covariance operator $C$ must satisfy $\int_{\mathcal{H}} \| x \|_{\mathcal{H}} d\mu(x) = \text{tr}(C) < \infty$ (Da Prato and Zabczyk, 2014). Mollification involves convolving white noise with a mollifier kernel  $k(s)>0$ that corresponds to a linear operator $T$, resulting in white noise that can be described by $\mathcal{N}(0, TT^*)$ (Higdon 2002). This alters the white noise to ensure that it lies within Hilbert space and satisfies the trace-class condition (Sec. 3.1). Without mollification, it is much more difficult to tune the noise hyperparameters, leading to the trained model being unable to generalise across different coordinate subsampling rates or different resolutions. We have clarified this in Section 3.2.
>
> > **Clarify what $v_0$ means**
>
> $v_0$, which lies in Banach space, is the input to the deep neural operator-based network. Each subsequent $v_l$ are the activations of the network, which also lie in Banach space, with $v_l \mapsto v_{l+1}$ being a single neural operator layer. We have clarified this in the updated paper.

---

> > ### Comment · Reviewer_gZH8 · 2023-11-21
> >
> > I thank the authors for their answers which serve as great clarification of their proposed model. I have raised my score by 1.

---

### Author Response · Authors · 2023-11-20
**Author Rebuttal**

We thank the reviewers for their time and constructive feedback which will improve our paper. In particular, for recognising that our proposal is “novel and significant”, “well designed”, and that “the results are convincing”.

We discuss here the point raised by two reviewers about the parallel works developing diffusion models in infinite dimensions, discussed in Section 6. These works, either available only on arXiv or published after May 28th (such as at NeurIPS 2023), are classified as contemporaneous according to ICLR guidelines and therefore do not necessitate direct comparison. It's noteworthy that our work, published on arXiv concurrently with these studies, is even cited by some as a concurrent work. Nevertheless, we have added additional references to these parallel works throughout the paper; we expanded the discussion section to elaborate on how these developments are distinct yet complementary to our work, acknowledged in the introduction that ours is not the sole approach in this area, and highlighted in the method section the greater theoretical treatment in these works in contrast to our focus on the design and practical scaling to complex high-resolution datasets.

---

### Meta-Review · Area_Chair_a3y7 · 2023-12-21

**Metareview:**

The reviewers have provided comprehensive feedback on the strengths and weaknesses of the paper titled "Diffusion Models in Infinite-Dimensional Spaces." The authors have responded to the feedback in a detailed manner, addressing many of the concerns raised.

The consensus among the reviewers is that the paper presents a novel and significant contribution to the field of generative modeling, particularly in the context of diffusion models operating in infinite-dimensional Hilbert spaces. The introduction of non-local integral operators, the development of a scalable architecture, and the impressive empirical results are highlighted as the primary strengths of the paper. The paper's clear presentation and the potential for the proposed model to achieve state-of-the-art results in generating high-resolution data have also been appreciated.

Regarding the weaknesses, the reviewers have raised concerns about the paper's initial framing of its contributions in relation to existing literature, the lack of detailed comparison with contemporaneous works, and certain imprecisions in the theoretical foundations presented in Section 3. In response, the authors have made revisions to acknowledge these contemporaneous works more thoroughly and to clarify the relative emphasis on practical implementation over theoretical treatment.

Given the revisions made by the authors and the discussion that has taken place, I recommend accepting this paper for publication. The practical contributions are substantial and represent a clear advancement in the application of infinite-dimensional diffusion models to complex and high-resolution datasets. The authors have made efforts to place their work within the context of the broader research landscape, thereby addressing the initial concerns about the framing of their contributions.

**Justification For Why Not Higher Score:**

While the paper has been significantly improved, it is important to note that there are still areas that could be enhanced in future work. The theoretical aspects of the proposed model could be further refined and clarified to match the depth of discussion found in some of the contemporaneous works. Additionally, the reviewers' suggestions for further experimental validation and discussion on computational costs and scaling limits are well-taken points that the authors could address in subsequent research. However, these points do not detract from the current contributions of the paper, which are sufficient to warrant acceptance.

**Justification For Why Not Lower Score:**

The acceptance score is justified as the paper makes a valuable contribution to the field, with practical implications that outweigh the initial concerns about the theoretical framing. The authors have successfully addressed many of the issues raised by the reviewers, and the paper presents empirical results that are convincing and of high quality. The architecture and methodology proposed by the authors are novel and represent an important step forward in the application of diffusion models to infinite-dimensional spaces.

---

### Decision · Program_Chairs · 2024-01-16

Accept (poster)